# Parameter Dynamics of Online Machine Learning and Test-time Adaptation

**Jae-Hong Lee**
Division of Language & AI
Hankuk University of Foreign Studies
Seoul, Republic of Korea
`ljh93ljh@hufs.ac.kr`

## Abstract

Pre-trained models based on deep neural networks hold strong potential for cross-domain adaptability. However, this potential is often impeded in online machine learning (OML) settings, where the breakdown of the independent and identically distributed (i.i.d.) assumption leads to unstable adaptation. While recent advances in test-time adaptation (TTA) have addressed aspects of this challenge under unsupervised learning, most existing methods focus exclusively on unsupervised objectives and overlook the risks posed by non-i.i.d. environments and the resulting dynamics of model parameters. In this work, we present a probabilistic framework that models the adaptation process using stochastic differential equations, enabling a principled analysis of parameter distribution dynamics over time. Within this framework, we find that the log-variance of the parameter transition distribution aligns closely with an inverse-gamma distribution under stable and high-performing adaptation conditions. Motivated by this insight, we propose Structured Inverse-Gamma Model Alignment (SIGMA), a novel algorithm that dynamically regulates parameter evolution to preserve inverse-gamma alignment throughout adaptation. Extensive experiments across diverse models, datasets, and adaptation scenarios show that SIGMA consistently enhances the performance of state-of-the-art TTA methods, highlighting the critical role of parameter dynamics in ensuring robust adaptation.

## 1 Introduction

The rapid advancement of deep neural networks (DNNs) has given rise to powerful pre-trained models capable of generalizing across a wide range of domains [12, 37, 56, 3, 4]. Despite their versatility, deploying these models in dynamic, resource-constrained environments remains challenging. This difficulty stems from their increasing computational demands and, more fundamentally, from the breakdown of the independent and identically distributed (i.i.d.) assumption in real-world online learning scenarios [16, 50]. Online machine learning (OML) [5] provides a paradigm for addressing such challenges by enabling models to adapt incrementally to non-stationary data streams, where full retraining on large, static datasets is impractical [18, 44]. In OML settings, data arrive sequentially from various domains, inherently violating the i.i.d. assumption that underpins conventional DNN training. This sequential domain shift often induces a severe degradation in model performance and generalization ability [21, 27, 19, 60, 9]. Consequently, a critical goal is to develop robust online adaptation methods to prevent collapse while preserving the adaptability under non-i.i.d. conditions.

Test-time adaptation (TTA) has recently emerged as a promising direction, combining the challenges of OML with unsupervised learning. TTA aims to adapt pre-trained models to unlabeled test data that arrive sequentially during deployment. The early TTA approach focused on unsupervised learning

39th Conference on Neural Information Processing Systems (NeurIPS 2025).

objectives. For example, TENT [61] replaces cross-entropy with entropy minimization over model predictions. While such methods offer improvements in early stages, they are prone to collapse when faced with extended domain shifts [6, 10, 62, 66, 51]. Recent efforts have addressed this instability by introducing sample-filtering strategies that downweight or discard high-uncertainty predictions during adaptation [45, 46, 29, 43]. Despite their empirical success, these methods remain narrowly focused on unsupervised objectives and do not explicitly address the fundamental instability introduced by the non-i.i.d. nature of the data and the evolving dynamics of model parameters.

In this work, we take a fundamentally different perspective. Rather than focusing solely on the loss function, we develop a probabilistic framework that explicitly models the dynamics of model parameters during adaptation. Our framework leverages stochastic differential equation (SDE) approximations of stochastic gradient descent (SGD) [33, 35], treating SGD as a continuous-time stochastic process. By discretizing the SDE, we derive a transition distribution that represents the evolution of the parameter uncertainty over time during the adaptation process (Section 3). Within this framework, we focus on the logarithmic variation (log-variance) of the transition distribution as a diagnostic signal. Empirical analysis under both i.i.d. and non-i.i.d. conditions, in both supervised (SL) and unsupervised (USL) settings, leads to three key findings (Section 4): (1) The log-variance distribution follows an inverse-gamma (IG) distribution in stable, high-performing adaptation scenarios. (2) Deviations from the IG distribution are strongly predictive of performance degradation. (3) State-of-the-art TTA methods tend to implicitly promote IG alignment, suggesting an unintentional but beneficial form of regularization.

These findings motivate our core hypothesis: explicitly maintaining IG alignment in the log-variance dynamics is critical for stable and effective adaptation in non-i.i.d. environments. To realize this principle, we introduce the Structured Inverse-Gamma Model Alignment (SIGMA) algorithm. SIGMA dynamically estimates an appropriate IG distribution using derivative-free optimization and adjusts the parameter update trajectory to preserve alignment throughout online adaptation. We validate SIGMA through extensive experiments across multiple models, datasets, and adaptation scenarios. Our results show that SIGMA consistently improves the performance of state-of-the-art TTA methods, supporting our central claim: alignment of parameter dynamics via IG regularization offers a robust and principled foundation under online adaptation settings.

## 2 Preliminaries

### 2.1 Test-Time Adaptation

TTA addresses the challenge of adapting a well-trained model, pretrained on large-scale labeled source data, to a stream of unlabeled test data from evolving domains. The objective is to adapt the model in real time to each new domain sample. Performance is measured by the average error rate (AER) across the test stream. TTA methods, developed from domain adaptation and continual learning research [10, 62, 66, 51], aim to handle changing data distributions. Recently, filtering strategies have been introduced to exclude or downweight uncertain predictions, improving robustness in multi-domain and non-i.i.d. environments [45, 46, 29, 43]. While most existing TTA methods focus on unsupervised loss design, our main contribution directly models and analyzes parameter changes during adaptation. Specifically, we propose a probabilistic framework that interprets TTA as an instance of online parameter evolution, offering a systematic method to understand the process.

### 2.2 The SDE Approximation

SGD lies at the heart of modern deep learning, and understanding its dynamics is fundamental to advancing theoretical and practical training aspects. A growing body of research has leveraged SDEs to approximate SGD from a continuous-time perspective, providing deeper insights into learning behavior [33, 35, 40, 1]. A key development is the use of stochastic modified equations to approximate discrete-time SGD updates with continuous-time SDEs [33]. This formulation captures both the deterministic gradient flow and stochastic fluctuations arising from mini-batch sampling, yielding a more complete description of the learning process. Later studies [35] have generalized this approach to various optimizers and used it to analyze the link between learning dynamics and generalization. This work builds upon this perspective and proposes a probabilistic framework that captures parameter dynamics from discrete-time observations in online adaptation.

# 3 Parameter Dynamics and Log-Variance Portraits

In this section, we present a theorem establishing a statistically robust measurement, the log-variance portrait, which tracks the evolution of the parameter distribution. Building on this, Section 4 reveals empirical patterns that characterize stable adaptation, and Section 5 applies these findings to develop our adaptation method.

## 3.1 Online Machine Learning Problem

We consider a DNN model $f : \mathcal{X} \to \mathcal{Y}$ parameterized by $\boldsymbol{w} \in \mathbb{R}^d$, which maps inputs $\boldsymbol{x} \in \mathcal{X}$ to labels $y \in \mathcal{Y}$. The model defines a conditional distribution $p(y|\boldsymbol{x}, \boldsymbol{w})$, used for inference. Let $\{(\boldsymbol{x}_n, y_n) \sim D_0 : n = 1 : N_0\}$ denote samples drawn from a source distribution $D_0$, used to pre-train a source model $f(\cdot; \hat{\boldsymbol{w}}_0)$. During online adaptation, the model encounters a stream of samples $z_k$ drawn from a sequence of target distributions $D_k \neq D_0$ at discrete time steps $t_k \in \{1, 2, 3, \ldots, K\}$. The adaptation objective is to minimize the expected risk:

$$G(\hat{\boldsymbol{w}}, t_k) = \mathbb{E}_{z_k \sim D_k}\left[\ell\left(f\left(\boldsymbol{x}_k; \hat{\boldsymbol{w}}(t_{k-1})\right)\right)\right], \tag{1}$$

where $z_k = (\boldsymbol{x}_k, y_k)$ in SL settings and $z_k = (\boldsymbol{x}_k)$ in USL settings. The loss function $\ell(\cdot)$ corresponds to cross-entropy in SL, and to entropy-based objectives in USL. At each time step, the model parameters are updated to minimize the risk:

$$\hat{\boldsymbol{w}}(t_k) = \arg\min_{\hat{\boldsymbol{w}}} G(\hat{\boldsymbol{w}}, t_k). \tag{2}$$

The adaptation process is implemented using an SGD-based optimizer, producing a time series of parameters $\{\hat{\boldsymbol{w}}(t_1), \hat{\boldsymbol{w}}(t_2), \ldots, \hat{\boldsymbol{w}}(t_k)\}$. However, this trajectory is vulnerable to degradation in non-i.i.d. environments or USL settings, where biased or noisy gradients can destabilize learning. Our goal is to characterize and improve this trajectory by building a probabilistic framework that captures the underlying stochasticity in the parameter evolution.

## 3.2 Probabilistic Framework for Parameter Dynamics

To model parameter evolution during adaptation, we approximate the discrete SGD updates using a continuous-time SDE following Li et al. [33]. When the learning rate $\eta$ is sufficiently small, the parameter updates can be approximated by:

$$d\boldsymbol{w}(t) = -g(\boldsymbol{w}, t)dt + \sqrt{\eta}\Sigma^{1/2}(\boldsymbol{w}, t)dW_t, \tag{3}$$

where $dW_t$ denotes a standard Brownian motion, $g(\boldsymbol{w}, t) = \nabla G(\boldsymbol{w}, t)$ denotes the gradient of the risk, and $\Sigma(\boldsymbol{w}, t)$ is the empirical covariance of the gradients. Specifically, we define: $\Sigma(\boldsymbol{w}, t) = 1/t \sum_{\tau=1}^{t} (g(\boldsymbol{w}, \tau) - \bar{g}(\boldsymbol{w}, \tau))(g(\boldsymbol{w}, \tau) - \bar{g}(\boldsymbol{w}, \tau))^{\top}$ with $\bar{g}(\boldsymbol{w}, t) = 1/t \sum_{\tau=1}^{t} g(\boldsymbol{w}, \tau)$ as the mean gradient. Following [33], we treat the entries of $\boldsymbol{w}$ as independent and approximate the full covariance matrix with a scalar multiple of the identity $\Sigma(\boldsymbol{w}, t) \approx \sigma_t^2 \mathbf{I}$, where $\sigma_t^2 = 1/d \, \mathrm{Tr}(\Sigma_t)$ and $\mathrm{Tr}(.)$ is the trace. The parameter distribution $p(\boldsymbol{w}(t))$ under this SDE evolves according to the Fokker–Planck–Kolmogorov (FPK) equation:

$$\frac{\partial p(\boldsymbol{w}(t))}{\partial t} = \sum_{i=1}^{d} \frac{\partial p(\boldsymbol{w}(t))}{\partial w_t^i}[g(\boldsymbol{w}, t)]_i + \frac{1}{2}\sum_{i=1}^{d}\sum_{j=1}^{d} \frac{\partial^2 p(\boldsymbol{w}(t))}{\partial w_t^i \partial w_t^j}\eta[\Sigma(\boldsymbol{w}, t)]_{ij}, \tag{4}$$

where $[\cdot]_i$ and $[\cdot]_{ij}$ denote vector and matrix components, respectively. The following theorem provides a tractable discrete-time approximation of the transition distribution implied by Eq. (3).

**Theorem 1** (Discretization of the SDE approximation). *Let $(t_{k-1}, t_k)$ be a sufficiently small discrete-time interval, and assume that both the gradient $g(\boldsymbol{w}, t_k)$ and the variance $\sigma_t^2$ remain approximately constant over this interval, denoted as $g_k$ and $\sigma_k^2$. Then, the transition distribution of the SDE can be approximated by:*

$$p(\boldsymbol{w}(t_k)|\boldsymbol{w}(t_{k-1})) \approx \mathcal{N}(\boldsymbol{w}(t_k)|\mu_{k|k-1}, \Sigma_{k|k-1}), \tag{5}$$

*where $\Delta t = t_k - t_{k-1} = \eta$, and the mean and covariance are given by:*

$$\mu_{k|k-1} = \boldsymbol{w}(t_{k-1}) - g_k\Delta t, \quad \Sigma_{k|k-1} = \sigma_k^2\Delta t^2\boldsymbol{I}, \tag{6}$$

*with $\sigma_k^2 = 1/d \, Tr[(g_k - \bar{g}_k)(g_k - \bar{g}_k)^{\top}], \bar{g}_k = 1/k \sum_{\tau=1}^{k} g_\tau$.*

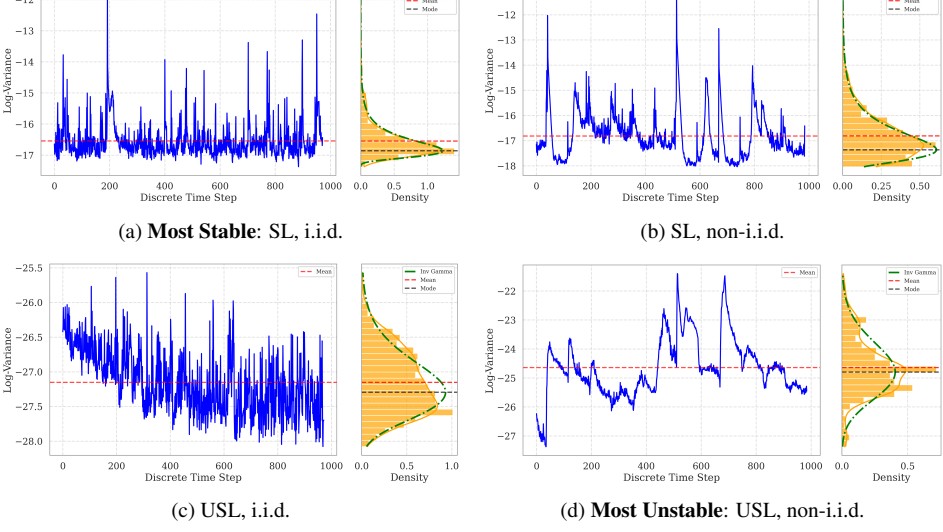

(a) **Most Stable**: SL, i.i.d.

(b) SL, non-i.i.d.

(c) USL, i.i.d.

(d) **Most Unstable**: USL, non-i.i.d.

Figure 1: Log-variance dynamics under SL and USL (TENT) settings in i.i.d. and non-i.i.d. environments. The blue line shows temporal evolution of log-variance. Orange boxes denote the empirical distribution of log-variance. The green dash-dot line represents the fitted IG distribution, with red and black dashed lines marking the IG mean and mode, respectively. The degradation in IG alignment reflects increasing instability across scenarios.

The proof is provided in Appendix A.1. This result allows us to use discrete-time gradient observations $g_k$ to approximate the evolution of the parameter distribution, which is originally defined in continuous time. Importantly, it reveals the variance term $\sigma_k^2 \Delta t^2$, often ignored in traditional SGD analysis. This variance shows local uncertainty and variability in the direction of the gradient. It is an important signal for assessing adaptation stability. While this variance term provides valuable insight, the raw variance $\sigma_k^2$ has extremely heavy tails and is difficult to analyze directly, even in a stable environment (See Appendix A.3). To address this, we instead focus on its logarithm: $v_k = \log(\sigma_k^2 \Delta t^2)$. The *log-variance portrait* refers to the empirical distribution obtained by collecting values of $v_k$ at each step during adaptation. This portrait illustrates the evolving distribution of local logarithmic variances in the gradients over time, allowing us to see fluctuations in gradient variability as adaptation progresses. By summarizing these local variances, the log-variance portrait offers a compact and interpretable way to visualize and quantify how parameter dynamics change throughout adaptation.

## 4 Relationship between Log-variance Portrait and Performance

We investigate how parameter dynamics evolve during adaptation by analyzing the behavior of the log-variance portrait across different online adaptation settings. Our goal is to determine whether the alignment between this distribution and an IG distribution is predictive of model performance, particularly in non-i.i.d. environments. Through extensive empirical analysis, we show that the goodness-of-fit (GoF) between the log-variance distribution and an IG distribution is a reliable indicator of adaptation success. In particular, maintaining high IG-GoF is essential for preventing performance degradation and enabling stable adaptation.

**Analysis Setup.** We evaluate model dynamics under both i.i.d. and non-i.i.d. environments, in both SL and USL settings. For USL, we adopt TENT [61] as a representative baseline. Experiments are conducted on ImageNet-C [21], which includes 15 corruption-based domains grouped into four broad categories. In the i.i.d. environment, domain samples are shuffled and presented randomly; in the non-i.i.d. environment, domains are presented sequentially in a fixed order. To quantify IG-GoF, we fit an IG distribution to the log-variance values using maximum likelihood estimation and assess fit using the Kolmogorov–Smirnov test. We report the resulting p-value as the alignment score. The performance improvement is measured using the relative average error rate (RAER), defined as: $\text{RAER} = (e_{\text{source}} - e_{\text{target}})/e_{\text{source}} * 100$, where $e_{\text{source}}$ is the AER of the source model and $e_{\text{target}}$ is the AER after adaptation. A negative RAER indicates performance degradation. Additional experimental details are provided in Section 6.1.

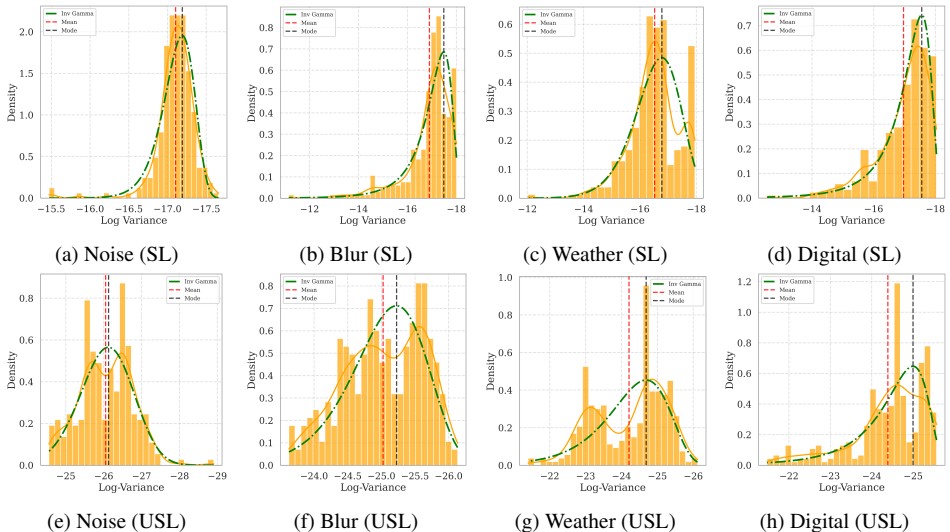

(a) Noise (SL)    (b) Blur (SL)    (c) Weather (SL)    (d) Digital (SL)

(e) Noise (USL)    (f) Blur (USL)    (g) Weather (USL)    (h) Digital (USL)

Figure 3: Log-variance portraits for each domain in a non-i.i.d. environment. The top row shows results under SL, and the bottom row under USL using TENT. The adaptation sequence follows four target domain (i.e., *Noise*, *Blur*, *Weather*, and *Digital*) with each domain visualized separately.

**Portraits Across All Domains.** We first investigate how the log-variance portrait evolves when varying one factor at a time from a stable baseline: SL under i.i.d. conditions. Figure 1 (a) shows that in this stable setting, the log-variance evolves smoothly over time and aligns closely with the IG distribution. This behavior generalizes across different model architectures and datasets (Appendix C.1 and C.6). When switching to a non-i.i.d. environment (Figure 1 (b)), the portrait partially deviates from the IG shape, indicating destabilization. A similar effect is seen when moving from SL to USL (Figure 1 (c)). When both non-i.i.d. input and unsupervised adaptation are combined, as in realistic TTA scenarios, Figure 1 (d) shows a significant breakdown in IG alignment, often yielding multimodal distributions. ***Takeaway 1.1: Both non-i.i.d. input streams and unsupervised objectives introduce instability into the adaptation process. IG alignment in the log-variance portrait is generally preserved only under stable learning conditions.***

**Performance Across All Domains.** We next examine how IG-GoF relates to adaptation performance. In Figure 2, lower p-values (weaker IG-GoF) consistently correspond to lower RAER (worse performance), while stronger alignment (higher p-values) correlates with performance improvements. Notably, the most severe degradation occurs under the combined non-i.i.d. and USL setting, highlighting the compounding effect of sequential domain shifts and lack of supervision. ***Takeaway 1.2: High IG-GoF is a strong predictor of adaptation success. The log-variance portrait thus provides a statistically grounded diagnostic signal for evaluating adaptation quality.*** These findings naturally lead to the next question: Can degradation in IG-GoF be traced to specific domains encountered during online adaptation?

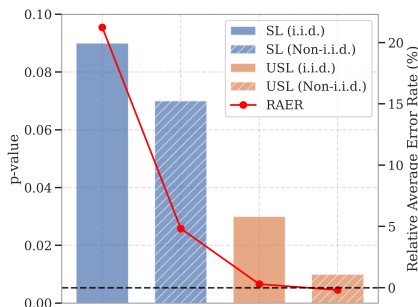

Figure 2: RAERs and corresponding p-values evaluated across all target domains for each adaptation setting.

**Portraits and Performance Across Individual Domains.** To localize adaptation instability, we examine IG-GoF across individual domain groups in ImageNet-C: *Noise*, *Blur*, *Weather*, and *Digital*. The order reflects the temporal sequence in which the model encounters each group during online adaptation. Figure 3 visualizes the log-variance portraits for each domain under SL (top row) and USL (bottom row) settings in the non-i.i.d. environment. In the SL setting, the portrait initially aligns well with the IG distribution in the *Noise* domain but progressively deteriorates, with a pronounced collapse in the *Weather* domain, indicating growing instability in parameter dynamics. Under USL, a similar trend is observed, with the *Weather* domain again showing the most severe deviation. Notably, the portrait becomes multimodal in this case, diverging from the unimodal structure characteristic of an IG distribution.

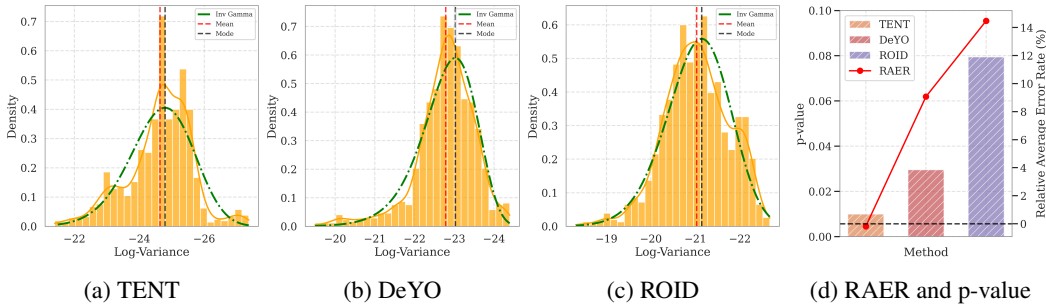

| (a) TENT | (b) DeYO | (c) ROID | (d) RAER and p-value |

Figure 5: Log-variance portraits (a–c) and corresponding RAER and p-value statistics (d) for different TTA methods under the USL setting in a non-i.i.d. environment. Subfigures (a–c) show the empirical log-variance distributions (orange boxes) and their alignment with the fitted IG distribution (green dash-dot line) for TENT, DeYO, and ROID, respectively. Subfigure (d) summarizes the RAER and IG-GoF (p-values) for each method.

Figure 4 quantitatively summarizes this trend, showing the RAER and p-values per domain. We observe that domains with p-values above approximately 0.01 exhibit positive RAER, indicating successful adaptation, whereas domains below this threshold experience performance degradation (RAER < 0), indicating performance degradation. ***Takeaway 2:*** *Maintaining sufficiently high IG-GoF is essential for avoiding performance degradation.* Deviation from the IG distribution is tightly coupled with adaptation failure, reinforcing the importance of preserving statistical structure in parameter dynamics throughout the adaptation process.

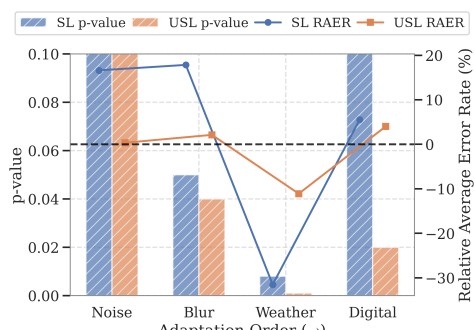

Figure 4: RAERs and corresponding p-values for each domain under a non-i.i.d. environment. Domains with p-values below the threshold ($\approx$0.01) tend to experience performance degradation.

**Portraits and Performance of Sample-Filtering Methods.** Recent sample-filtering TTA methods have demonstrated strong empirical success in mitigating the effects of non-i.i.d. environment. To better understand this success, we examine how these methods influence the IG-GoF of the log-variance distribution. Figure 5 (a–c) compares the log-variance portraits of the baseline method TENT with two modern sample-filtering approaches: DeYO [29] and ROID [43]. Unlike TENT, which shows apparent deviation from the IG form, both DeYO and ROID yield distributions that are more closely aligned with the IG shape, indicating greater statistical stability. Figure 5 (d) reports RAERs and the corresponding p-values for each method. Both DeYO and ROID achieve substantially higher p-values and improved RAERs compared to TENT, confirming that stronger IG-GoF is associated with better adaptation performance. ***Takeaway 3:*** *Sample-filtering methods implicitly encourage IG alignment in parameter dynamics.* This alignment underlie their robustness in challenging non-i.i.d. environments. Nevertheless, ROID, despite its relatively strong IG-GoF, still exhibits minor deviations from the ideal IG distribution. This gap suggests that adaptation performance could be further enhanced by explicitly regulating parameter dynamics to maintain IG alignment throughout the learning process, rather than relying on implicit regularization effects.

## 5 Structured Inverse-Gamma Model Alignment

We introduce SIGMA, a principled algorithm designed to correct degraded parameter trajectories under USL or non-i.i.d. conditions, thereby enabling stable and high-performing adaptation. Building on our empirical insights from the previous section, SIGMA explicitly regulates the evolution of model parameters to preserve statistical alignment with the IG distribution, a property closely associated with successful adaptation. In the remainder of this section, we describe the algorithm's components in detail and analyze how it dynamically modulates parameter updates through time-varying adjustment of the update interval.

## 5.1 Algorithm

As established in the previous section, a high GoF between the log-variance portrait and the IG distribution is a key indicator of stable adaptation. Based on this observation, we hypothesize that adaptation can be improved by (a) identifying an appropriate IG distribution in real time, and (b) actively regulating the parameter dynamics to preserve alignment with it throughout the learning process. SIGMA implements this principle via three core steps: (1) *Estimate Step*: online estimation of an IG distribution that best fits the observed log-variance dynamics; (2) *Align Step*: computation of correction coefficients to align the current log-variance with the estimated IG distribution; (3) *Conjugate Step*: modification of the parameter update based on the aligned distribution. These steps collectively produce a dynamic rescaling of the effective update interval $\Delta t$ at each time step, thereby inducing a statistically calibrated trajectory for parameter adaptation. We begin by reparameterizing the log-variance $v_k$ at each time step using two correction coefficients: a location shift $c_k \leq 0$ and a scale factor $b_k > 1$. The corrected log-variance $\tilde{v}_k$ is defined as:

$$\tilde{v}_k = (v_k - c_k)/b_k + 2 \log \lambda, \tag{7}$$

where alignment strength $\lambda$ quantifies how closely the updated distribution should follow the real-time reference distribution, which is assumed to follow the IG form.

**Estimate Step.** We aim to identify an distribution $\text{IG}(v; \alpha_{k-1}, 1)$ that best explains the history of calibrated log-variances $\tilde{v}_{1:k-1}$. This strategy is framed as a KL divergence minimization:

$$q(v|\tilde{v}_{1:k-1}; \alpha_{k-1}) = \underset{q \in \text{IG}}{\arg \min} \, D_{\mathbb{KL}}(p(v|\tilde{v}_{1:k-1}) \,||\, q(v|\tilde{v}_{1:k-1}; \alpha)). \tag{8}$$

We solve this optimization using the Nelder–Mead simplex method [14], a derivative-free algorithm well-suited for scalar-valued observations.

**Align Step.** Given the estimated IG distribution, we compute optimal correction coefficients $(c_k^*, b_k^*)$ that align the current log-variance $v_k$ with the target distribution by the simplex method. This optimization is formulated as the negative log-likelihood minimization:

$$c_k^*, b_k^* = \underset{(c_k, b_k)}{\arg \min} - \log q(\tilde{v}_k|\tilde{v}_{1:k-1}; \alpha_{k-1}). \tag{9}$$

We impose constraints $c_k^* \leq 0$ and $b_k^* > 1$ to ensure stable reparameterization, and clip the solution if it falls outside the valid region. Applying these coefficients, the corrected log-variance becomes:

$$\tilde{v}_k = 2 \log |\sigma_k \underbrace{(\lambda r_k T_k) \Delta t}_{\Delta t_k}| = 2 \log |\sigma_k \Delta t_k|, \tag{10}$$

where the adjusted time interval is defined as $\Delta t_k = (\lambda r_k T_k) \Delta t$, with $r_k = \exp(-c_k^*/2b_k^*)$ and $T_k = (\sigma_k \Delta t)^{(1-b_k^*)/b_k^*}$. The derivation is detailed in Appendix B.2. This yields the modified transition distribution:

$$p^*(\boldsymbol{w}(t_k)|\boldsymbol{w}(t_{k-1})) = \mathcal{N}(\boldsymbol{w}(t_k)|\mu_{k|k-1}, \Sigma_{k|k-1}), \tag{11}$$

where

$$\mu_{k|k-1} = \boldsymbol{w}(t_{k-1}) - g_k \Delta t_k, \quad \Sigma_{k|k-1} = \sigma_k^2 \Delta t_k^2 \mathbf{I}. \tag{12}$$

The effective update interval $\Delta t_k$ is shaped by two opposing influences: (1) $r_k > 1$, which enlarges $\Delta t_k$ to correct location misalignment; and (2) $T_k < 1$, which reduces $\Delta t_k$ in response to high gradient variance. Together, these terms adjust the parameter trajectory to preserve IG conformity and enhance adaptation stability.

**Conjugate Step.** We integrate the corrected dynamics into the parameter trajectory until $t_k$. Using the modified transition distribution, we recursively calculate the marginal distribution as follows:

$$p(\boldsymbol{w}(t_k)) = \int p^*(\boldsymbol{w}(t_k)|\boldsymbol{w}(t_{k-1})) p(\boldsymbol{w}(t_{k-1})) \mathrm{d}\boldsymbol{w}(t_{k-1}) = \mathcal{N}(\boldsymbol{w}(t_k)|\mu_k, \Sigma_k) \tag{13}$$

where $\mu_k = \hat{\boldsymbol{w}}_0 - \sum_{i=1}^{k} g_i \Delta t_i$ and $\Sigma_k = \Sigma_0 + \sum_{i=1}^{k} \sigma_i^2 \Delta t_i^2 \mathbf{I}$ with $\Sigma_0 = \mathbf{0}$ and initial prior $p(\boldsymbol{w}(t_0)) = \mathcal{N}(\hat{\boldsymbol{w}}_0, \Sigma_0)$. This resulting distribution over parameters is considered a conjugate prior for the predictive distribution:

$$p(y|\boldsymbol{x}_k) = \int p(y|\boldsymbol{x}_k, \boldsymbol{w}(t_k)) p(\boldsymbol{w}(t_k)) \mathrm{d}\boldsymbol{w}(t_k). \tag{14}$$

To proceed, we apply the plug-in approximation [44], we evaluate the integral at the posterior mean $\mu_k$, yielding the final prediction $f(\boldsymbol{x}_k; \mu_k)$ used for risk computation in Eq. (1). The algorithm then proceeds to the next time step $t_{k+1}$, iterating the Estimate–Align–Conjugate cycle. Illustrations of the algorithm and pseudocode are provided in Appendix B.

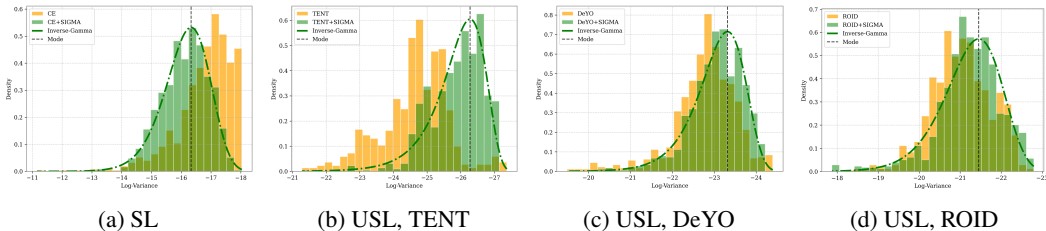

| (a) SL | (b) USL, TENT | (c) USL, DeYO | (d) USL, ROID |

Figure 6: Log-variance portraits under non-i.i.d. environments for SL and USL settings, before (orange) and after (green) applying SIGMA. Green distributions exhibit closer alignment with the IG reference, indicating improved parameter dynamics.

## 5.2 Effectiveness

We evaluate SIGMA in both SL and USL settings under non-i.i.d. environments. For the USL case, we integrate SIGMA with three representative TTA methods: TENT, DeYO, and ROID. Figure 6 illustrates the qualitative effect of SIGMA on the log-variance portraits. In all cases, SIGMA reshapes the distribution to more closely follow the IG structure, validating its mechanism of action. Even robust baselines such as ROID benefit from additional stabilization when augmented with SIGMA. Figure 7 shows the quantitative improvements in both RAER and IG-GoF (p-values). Across all tested methods and conditions, SIGMA consistently increases the p-value, indicating improved statistical alignment with the IG distribution. This alignment is invariably accompanied by a corresponding gain in RAER, demonstrating that

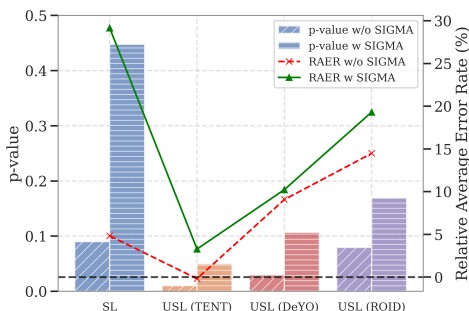

Figure 7: RAERs and corresponding p-values, before (slash-pattern bars) and after (vertical line-pattern bars) applying SIGMA.

better IG-GoF leads directly to improved adaptation performance. These results support our central hypothesis: preserving alignment with the IG distribution is key to preventing collapse and enabling robust online adaptation. SIGMA functions as a distribution-aware mechanism that enhances both statistical regularity and practical performance across diverse adaptation settings.

# 6 Comparison with State-of-the-art TTA Methods

In the previous sections, we demonstrated the effectiveness of SIGMA across a range of settings. To further validate our approach, we follow the standardized TTA benchmark protocol [42] under various scenarios. All reported results are averaged over four random seeds, and both AERs and corresponding standard deviations are presented to ensure statistical reliability and reproducibility. All experiments were conducted using a single NVIDIA GeForce RTX 4090 GPU.

## 6.1 Experimental Details

**Datasets** We evaluated SIGMA across both multi-domain and single-domain datasets. For multi-domain adaptation, we used ImageNet-C and D109; for single-domain evaluation, we used Rendition [22] and Sketch [21]. Multi-domain adaptation on ImageNet-C is the default dataset; other datasets are used in Appendix C. ImageNet-C extends the original ImageNet dataset, consisting of 1,281,167 training images and 50,000 test images, by applying 15 types of corruption (e.g., Gaussian noise, shot noise, defocus blur, frost, JPEG compression) at five severity levels. Following standard practice [45, 29, 43], we used severity level 5 and treated each corruption type as a distinct domain. The default domain order is Noise, Blur, Weather, and Digital. D109, derived from DomainNet [48], consists of five natural domains (i.e., clipart, infograph, painting, real, and sketch) and contains 109 classes overlapping with ImageNet. This dataset enabled evaluation under real-world domain shifts. To assess adaptation in single-domain settings, we adopted: Rendition, which comprises 30,000 stylized renderings of 200 ImageNet classes curated via Amazon Mechanical Turk; and Sketch, which contains 50,000 black-and-white sketches across 1,000 ImageNet classes collected via Google Image Search.

Table 1: AERs (%) and corresponding standard deviations in Correlated Input on ImageNet-C. The **bold** number indicates the best result.

| Method | Noise | | | Blur | | | | Weather | | | | Digital | | | | AER |
|---|---|---|---|---|---|---|---|---|---|---|---|---|---|---|---|---|
| | gaussian | shot | impulse | defocus | glass | motion | zoom | snow | frost | fog | bright | contrast | elastic | pixelate | jpeg | |
| Source | 43.9 | 43.3 | 43.4 | 69.7 | 78.3 | 59.6 | 69.1 | 40.1 | 44.3 | 36.3 | 26.5 | 50.6 | 67.6 | 60.6 | 43.4 | 51.8 |
| RoTTA | 43.9 | 43.3 | 43.3 | 69.7 | 77.8 | 59.4 | 68.7 | 39.8 | 42.5 | 35.9 | 26.2 | 49.8 | 66.6 | 60.2 | 43.4 | 51.4±0.02 |
| SAR | 44.1 | 43.8 | 43.7 | 69.7 | 77.3 | 57.1 | 66.8 | 41.3 | 41.5 | 42.3 | 26.3 | 50.2 | 64.1 | 57.2 | 42.0 | 51.2±0.30 |
| TENT | 43.9 | 43.1 | 43.3 | 69.4 | 76.5 | 57.9 | 67.1 | 40.5 | 44.7 | 51.1 | 27.3 | 47.1 | 64.5 | 58.6 | 43.2 | 51.9±0.08 |
| +SIGMA | 43.8 | 42.9 | 43.0 | 67.9 | 73.3 | 56.2 | 64.8 | 39.8 | 56.8 | 37.8 | 26.1 | 43.5 | 68.1 | 49.6 | 38.5 | 50.1±0.49 |
| EATA | 44.3 | 43.8 | 43.5 | 69.0 | 74.3 | 56.7 | 64.2 | 39.9 | 44.7 | 46.2 | 25.7 | 45.4 | 61.2 | 54.0 | 40.6 | 50.2±0.09 |
| +SIGMA | 43.7 | 42.6 | 42.1 | 63.8 | 64.7 | 51.8 | 57.8 | 37.5 | 37.1 | 32.3 | 23.9 | 40.8 | 53.8 | 44.7 | 35.9 | 44.8±0.04 |
| DeYO | 43.6 | 41.9 | 40.7 | 66.3 | 70.3 | 54.7 | 62.7 | 38.3 | 38.7 | 36.9 | 24.6 | 43.1 | 55.0 | 50.1 | 39.2 | 47.1±0.13 |
| +SIGMA | 43.6 | 42.3 | 41.8 | 65.4 | 68.1 | 53.5 | 65.3 | 39.3 | 37.9 | 32.2 | 24.6 | 41.2 | 58.4 | 46.2 | 36.9 | 46.5±0.23 |
| ROID | 42.8 | 40.5 | 39.8 | 63.0 | 63.3 | 49.8 | 56.6 | 36.8 | 36.3 | 31.4 | 24.8 | 39.6 | 56.6 | 47.2 | 36.5 | 44.3±0.14 |
| +SIGMA | **42.7** | **40.3** | **39.6** | **61.1** | **57.0** | **46.9** | **51.8** | **36.1** | **34.4** | **30.5** | **23.1** | **38.3** | **49.1** | **43.1** | **33.0** | **41.8±0.14** |

**Scenarios.** We considered two representative unsupervised online adaptation scenarios (i.e., continual TTA setup), each simulating different forms of real-world distribution shift. In the *Correlated Input* setting, domain-wise data were presented to the model in a fixed sequence, simulating temporally evolving input distributions [6, 62, 66]. In the *Correlated Label* setting, samples were drawn from localized label distributions generated by a Dirichlet distribution with concentration parameter $\gamma$ [17, 46, 68]. Lower values of $\gamma$ produced stronger label locality and non-stationarity, while $\gamma = \infty$ reduced to the Correlated Input scenario.

**Methods.** We compared SIGMA-augmented methods against a range of representative TTA baselines: TENT [61] minimizes entropy loss using the model's own predictions to reduce uncertainty during adaptation. RoTTA [66] employs a student-teacher architecture with cross-entropy loss and data augmentation, training a student model to adapt while maintaining alignment with a stable teacher. SAR [46] integrates sharpness-aware minimization to avoid sharp local optima and resets the model to the source checkpoint when loss exceeds a predefined threshold. EATA [45] extends TENT by filtering out high-entropy (i.e., low-confidence) samples based on a fixed threshold to prevent collapse. DeYO [29] identifies reliable samples using entropy and pseudo-label consistency under object-destructive transformations. ROID [43] regularizes entropy to account for label distribution diversity and filters low-confidence samples during training. To ensure a fair comparison, we disabled ROID's optional prior correction module in all experiments, as it is orthogonal and applicable to other methods.

**Implementation Details.** Following prior works [45, 29, 43], we used the base version of Vision Transformer (ViT) [12] with the self-supervised D2V model [4] as our default backbone. We also evaluated SIGMA using Swin Transformer (Swin) [37] to assess architectural generalization. All source models were initialized using publicly available weights pre-trained on ImageNet to ensure fair comparison and reproducibility. Consistent with previous studies [34, 45, 46, 43, 31], we restricted training to normalization layers, either batch normalization [24] or layer normalization [2], depending on the model architecture. We adopted official implementations and hyperparameters from each method's original paper and the standardized TTA benchmark suite [42]. When method-specific settings were unavailable for a dataset or model, we defaulted to the best-performing configuration reported for ROID. We used SGD with a momentum of 0.9 with the source-parameter averaging [46, 43, 31] and a batch size of 64 as the base optimizer. Learning rates were set to $1.0 \times 10^{-5}$ for D2V, $2.5 \times 10^{-4}$ for ViT and Swin, and $1.0 \times 10^{-3}$ for SAR (using the SAM optimizer [13]) across both ViT and Swin models. The SIGMA alignment strength $\lambda$ was fixed per method and held constant across experiments unless otherwise noted: $\lambda = 5.0 \times 10^{-5}$ for TENT, $7.5 \times 10^{-5}$ for EATA and ROID, and $5.0 \times 10^{-5}$ for DeYO. We implemented the Kolmogorov–Smirnov test via the `scipy.stats.kstest` and the Nelder–Mead simplex method via the `scipy.stats.fit`.

## 6.2 Results

**Correlated Input.** Table 1 summarizes the results on the ImageNet-C dataset under the Correlated Input setting, comparing baseline TTA methods with and without SIGMA. Among the baselines, sample-filtering approaches such as EATA, DeYO, and ROID outperformed other strategies, including student–teacher frameworks (i.e., RoTTA [66]) and sharpness-aware minimization (i.e., SAR [46]). ROID achieved the highest baseline accuracy. When we applied SIGMA, all methods showed further improvements. Specifically, TENT, EATA, DeYO, and ROID improved by 1.8%, 5.4%, 0.6%, and 2.5%, respectively. These results demonstrate SIGMA's ability to enhance stability by regulating parameter dynamics via IG alignment.

Table 2: AERs (%) and standard deviations in Correlated Label ($\gamma = 0.1$) on ImageNet-C. The **bold** number indicates the best result.

| Method | Noise | | | Blur | | | | Weather | | | | Digital | | | | AER |
|---|---|---|---|---|---|---|---|---|---|---|---|---|---|---|---|---|
| | gaussian | shot | impulse | defocus | glass | motion | zoom | snow | frost | fog | bright | contrast | elastic | pixelate | jpeg | |
| Source | 43.9 | 43.3 | 43.4 | 69.7 | 78.3 | 59.6 | 69.1 | 40.1 | 44.3 | 36.3 | 26.5 | 50.6 | 67.6 | 60.6 | 43.4 | 51.8 |
| RoTTA | 43.5 | 41.2 | 40.8 | 68.4 | 71.1 | 56.3 | 64.4 | 39.1 | 38.3 | 38.6 | 28.3 | 65.4 | 67.5 | 67.4 | 49.4 | 52.0±0.06 |
| SAR | 43.9 | 41.7 | 40.9 | 68.4 | 71.8 | 55.0 | 63.4 | 39.3 | 39.1 | 38.8 | 25.3 | 44.8 | 58.0 | 49.9 | 39.3 | 48.0±0.10 |
| TENT | 44.0 | 43.5 | 43.8 | 70.8 | 78.3 | 59.9 | 68.8 | 42.4 | 52.0 | 56.5 | 30.2 | 64.7 | 68.7 | 63.2 | 44.7 | 55.4±1.58 |
| +SIGMA | 43.4 | 41.8 | 41.6 | 66.9 | 75.4 | 56.0 | 68.5 | 41.6 | 60.9 | 33.3 | 25.4 | 43.0 | 69.3 | 50.1 | 39.6 | 50.5±0.16 |
| EATA | 43.5 | 40.5 | 39.6 | 61.7 | 62.0 | 48.1 | 56.0 | 36.7 | 36.0 | 32.9 | 23.0 | 37.1 | 53.2 | 44.5 | 34.7 | 43.3±0.03 |
| +SIGMA | 41.4 | 39.3 | 39.2 | 56.9 | 57.0 | 47.6 | 53.6 | 35.3 | 34.8 | 30.5 | 22.7 | 38.1 | 45.3 | 38.3 | 32.8 | 40.9±0.06 |
| DeYO | 41.3 | 38.8 | 38.8 | 60.8 | 61.0 | 52.3 | 70.6 | 42.5 | 40.3 | 40.8 | 26.1 | 64.3 | 66.4 | 48.1 | 42.9 | 49.0±2.83 |
| +SIGMA | 41.6 | 39.7 | 39.8 | 60.7 | 68.9 | 49.9 | 80.4 | 37.1 | 35.5 | 30.9 | 23.6 | 38.2 | 51.4 | 40.4 | 34.8 | 44.9±0.26 |
| ROID | 40.6 | 39.4 | 39.3 | 54.8 | 55.4 | 46.4 | 53.1 | 35.5 | 34.7 | 30.0 | 23.7 | 36.1 | 48.0 | 41.4 | 34.9 | 40.9±0.10 |
| +SIGMA | **40.2** | **38.4** | **38.4** | **51.6** | **49.5** | **42.2** | **46.9** | **33.1** | **32.8** | **28.8** | **22.0** | **35.5** | **39.7** | **35.5** | **30.9** | **37.7±0.01** |

Table 3: AERs (%) and standard deviations in Correlated Label ($\gamma = 0.0$) on ImageNet-C. The **bold** number indicates the best result.

| Method | Noise | | | Blur | | | | Weather | | | | Digital | | | | AER |
|---|---|---|---|---|---|---|---|---|---|---|---|---|---|---|---|---|
| | gaussian | shot | impulse | defocus | glass | motion | zoom | snow | frost | fog | bright | contrast | elastic | pixelate | jpeg | |
| Source | 43.9 | 43.3 | 43.4 | 69.7 | 78.3 | 59.6 | 69.1 | 40.1 | 44.3 | 36.3 | 26.5 | 50.6 | 67.6 | 60.6 | 43.4 | 51.8 |
| RoTTA | 43.8 | 42.0 | 42.0 | 69.9 | 74.5 | 59.3 | 67.4 | 40.3 | 39.5 | 40.2 | 29.0 | 74.5 | 72.4 | 72.8 | 51.5 | 54.6±0.04 |
| SAR | 44.2 | 41.8 | 41.0 | 67.6 | 71.7 | 54.8 | 63.5 | 39.2 | 39.0 | 38.2 | 25.6 | 67.5 | 66.0 | 57.9 | 39.0 | 50.5±1.38 |
| TENT | 44.1 | 44.0 | 44.0 | 71.1 | 79.2 | 61.6 | 69.8 | 43.2 | 53.1 | 55.9 | 30.8 | 48.7 | 69.4 | 69.1 | 58.9 | 56.2±0.98 |
| +SIGMA | 43.4 | 41.8 | 41.7 | 67.5 | 75.3 | 56.3 | 67.2 | 40.9 | 58.6 | 33.9 | 25.4 | 43.1 | 66.7 | 50.1 | 39.8 | 50.1±0.01 |
| EATA | 43.5 | 42.4 | 42.4 | 65.9 | 71.9 | 55.9 | 64.0 | 40.5 | 46.5 | 40.9 | 25.6 | 43.1 | 63.4 | 51.4 | 39.9 | 49.1±0.34 |
| +SIGMA | 42.9 | 42.4 | 42.8 | 70.1 | 71.2 | 57.5 | 64.9 | 36.8 | 41.7 | 36.8 | 26.4 | 58.7 | 52.6 | 47.3 | 38.6 | 48.9±0.17 |
| DeYO | 41.5 | 38.9 | **38.9** | 61.7 | 61.3 | 51.8 | 72.0 | 42.2 | 41.6 | 39.7 | 26.5 | 56.4 | 57.1 | 47.3 | 41.4 | 47.9±0.57 |
| +SIGMA | 41.7 | 39.9 | 40.0 | 61.2 | 66.7 | 50.8 | 82.3 | 37.4 | 35.8 | 31.2 | 23.7 | 39.5 | 51.6 | 40.8 | 35.6 | 45.2±0.19 |
| ROID | 41.1 | 39.5 | 39.6 | 58.5 | 57.6 | 47.8 | 55.7 | 35.9 | 35.3 | 30.6 | 23.9 | 38.0 | 48.9 | 42.0 | 35.6 | 42.0±0.15 |
| +SIGMA | **40.8** | **39.3** | 39.5 | **55.6** | **52.8** | **44.9** | **51.7** | **34.5** | **34.3** | **29.9** | **22.9** | **37.3** | **43.4** | **38.1** | **32.7** | **39.8±0.05** |

**Correlated Label.** We then evaluated SIGMA under more challenging scenarios involving shifts in label distribution. Table 2 and Table 3 present the results for $\gamma = 0.1$ and $\gamma = 0.0$, which induced increasingly localized label shifts. These settings exacerbated the risk of performance degradation, particularly for methods such as TENT. Despite this, SIGMA consistently prevented degradation. Despite this, SIGMA consistently prevented degradation. TENT's performance improved by 6.1% and 4.9% under the two $\gamma$ settings, effectively reversing collapse. ROID also benefited from improvements of 2.2% and 3.2%, reinforcing the method's broad applicability.

**Additional Results.** We extended our experiments to test generalizability for diverse model architectures, realistic scenarios, and datasets. The results in Appendix C showed that SIGMA consistently improved performance across various experimental settings. These results supported our central claim: enforcing IG alignment in parameter dynamics prevented collapse and led to consistent performance improvements under online adaptation settings.

# 7 Conclusion

In this work, we proposed a probabilistic framework to model the dynamics of model parameters during online adaptation using an SDE formulation. By discretizing the SDE, we derived a transition distribution that captured the time-evolving behavior of parameters throughout the adaptation process. Our empirical analysis revealed that the log-variance of this transition distribution served as a key indicator of adaptation stability. We found that successful adaptation consistently coincided with strong alignment between the log-variance portrait and the IG distribution. When this alignment broke down, models experienced performance degradation or collapse. Motivated by these findings, we introduced SIGMA, an algorithm that dynamically estimated the target IG distribution and regularized parameter updates to preserve alignment. SIGMA achieved this by adaptively rescaling the update interval based on variance dynamics. When applied to state-of-the-art TTA methods, SIGMA consistently improved performance across architectures, datasets, and realistic scenarios.

**Acknowledgement** This work was supported by the Ministry of Education of the Republic of Korea and the National Research Foundation of Korea (NRF-2025S1A5C3A01008166)

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

# A   Parameter Dynamics and Log-Variance Portraits

In Appendix A.1, we present the proof of Theorem 1, which characterizes the time-varying parameter distribution during adaptation. Building on this theorem, in Appendix A.3, we compare the dynamics of the raw variance and its logarithmic counterpart (i.e., log-variance). Our empirical observations show that the log-variance offers a more stationary and statistically tractable distribution, enabling practical computation of distributional alignment during adaptation.

## A.1   Proof of Theorem 1

*Proof.* Following Theorem 5.4 in Särkkä & Solin [53], the evolution of the mean $\boldsymbol{m}(t) = \mathbb{E}[\boldsymbol{w}(t)]$ and the covariance $\boldsymbol{S}(t) = \mathrm{Cov}[\boldsymbol{w}(t)]$ for the FPK solution $p(\boldsymbol{w}, t)$ from Eq. (4) is given by:

$$\frac{\mathrm{d}\boldsymbol{m}}{\mathrm{d}t} = -\int g(\boldsymbol{w}, t)p(\boldsymbol{w}, t)\mathrm{d}\boldsymbol{w},$$

$$\frac{\mathrm{d}\boldsymbol{S}}{\mathrm{d}t} = -\int g(\boldsymbol{w}, t)(\boldsymbol{w} - \boldsymbol{m})^\top p(\boldsymbol{w}, t)\mathrm{d}\boldsymbol{w}$$
$$- \int (\boldsymbol{w} - \boldsymbol{m})g^\top(\boldsymbol{w}, t)p(\boldsymbol{w}, t)\mathrm{d}\boldsymbol{w}$$
$$+ \int L(\boldsymbol{w}, t)L^\top(\boldsymbol{w}, t)p(\boldsymbol{w}, t)\mathrm{d}\boldsymbol{w},$$

where $L(\boldsymbol{w}, t) = \sqrt{\eta}\sigma_t\mathbf{I}$. Assuming a Gaussian approximation to the parameter distribution [28], we write:

$$p(\boldsymbol{w}, t) \approx \mathcal{N}(\boldsymbol{w}|\boldsymbol{m}(t), \boldsymbol{S}(t)).$$

We linearize $g(\boldsymbol{w}, t)$ and $L(\boldsymbol{w}, t)$ around $\boldsymbol{m}(t)$ via first-order Taylor expansion:

$$g(\boldsymbol{w}, t) \approx g(\boldsymbol{m}, t) + \mathcal{G}(\boldsymbol{m}, t)(\boldsymbol{w} - \boldsymbol{m}),$$
$$L(\boldsymbol{w}, t) \approx L(\boldsymbol{m}, t),$$

where $\mathcal{G}(\boldsymbol{m}, t)$ is the Jacobian of $g$ evaluated at $\boldsymbol{m}$. Substituting these into the FPK evolution yields:

**Mean:**

$$\frac{\mathrm{d}\boldsymbol{m}}{\mathrm{d}t} = -\int [g(\boldsymbol{m}, t) + \mathcal{G}(\boldsymbol{m}, t)(\boldsymbol{w} - \boldsymbol{m})]\mathcal{N}\mathrm{d}\boldsymbol{w}$$
$$= -g(\boldsymbol{m}, t) - \mathcal{G}(\boldsymbol{m}, t)\underbrace{\int (\boldsymbol{w} - \boldsymbol{m})\mathcal{N}\mathrm{d}\boldsymbol{w}}_{0}$$
$$= -g(\boldsymbol{m}, t).$$

**Covariance:**

$$\frac{\mathrm{d}\boldsymbol{S}}{\mathrm{d}t} = -\int [g(\boldsymbol{m}, t) + \mathcal{G}(\boldsymbol{m}, t)(\boldsymbol{w} - \boldsymbol{m})](\boldsymbol{w} - \boldsymbol{m})^\top \mathcal{N}\mathrm{d}\boldsymbol{w}$$
$$- \int (\boldsymbol{w} - \boldsymbol{m})[g(\boldsymbol{m}, t) + \mathcal{G}(\boldsymbol{m}, t)(\boldsymbol{w} - \boldsymbol{m})]^\top \mathcal{N}\mathrm{d}\boldsymbol{w}$$
$$+ \int L(\boldsymbol{m}, t)L^\top(\boldsymbol{m}, t)\mathcal{N}\mathrm{d}\boldsymbol{w}.$$

We compute each term separately:

$$-\int [g(\boldsymbol{m}, t) + \mathcal{G}(\boldsymbol{m}, t)(\boldsymbol{w} - \boldsymbol{m})r](\boldsymbol{w} - \boldsymbol{m})^\top \mathcal{N}\mathrm{d}\boldsymbol{w} = -g(\boldsymbol{m}, t)\underbrace{\int (\boldsymbol{w} - \boldsymbol{m})^\top \mathcal{N}\mathrm{d}\boldsymbol{w}}_{0}$$
$$- \mathcal{G}(\boldsymbol{m}, t)\underbrace{\int (\boldsymbol{w} - \boldsymbol{m})(\boldsymbol{w} - \boldsymbol{m})^\top \mathcal{N}\mathrm{d}\boldsymbol{w}}_{S}$$
$$= -\mathcal{G}(\boldsymbol{m}, t)\boldsymbol{S}. \qquad\qquad (a)$$

$$-\int (\boldsymbol{w} - \boldsymbol{m})[g(\boldsymbol{m},t) + \mathcal{G}(\boldsymbol{m},t)(\boldsymbol{w} - \boldsymbol{m})r]^\top \mathcal{N}\mathrm{d}\boldsymbol{w} = -\underbrace{\int (\boldsymbol{w} - \boldsymbol{m})\mathcal{N}\mathrm{d}\boldsymbol{w}}_{0}\, g(\boldsymbol{m},t)^\top$$

$$-\underbrace{\int (\boldsymbol{w} - \boldsymbol{m})(\boldsymbol{w} - \boldsymbol{m})^\top \mathcal{N}\mathrm{d}\boldsymbol{w}}_{\boldsymbol{S}}\, \mathcal{G}^\top(\boldsymbol{m},t)$$

$$= -\boldsymbol{S}\mathcal{G}^\top(\boldsymbol{m},t). \tag{b}$$

$$\int L(\boldsymbol{m},t)L^\top(\boldsymbol{m},t)\mathcal{N}\mathrm{d}\boldsymbol{w} = L(\boldsymbol{m},t)L^\top(\boldsymbol{m},t)$$

$$= \eta\sigma_t^2\mathbf{I}. \tag{c}$$

Summing Eqs. (a-c), we obtain the covariance dynamics:

$$\frac{\mathrm{d}\boldsymbol{S}}{\mathrm{d}t} = -\boldsymbol{S}\mathcal{G}^\top(\boldsymbol{m},t) - \mathcal{G}(\boldsymbol{m},t)\boldsymbol{S} + \eta\sigma_t^2\mathbf{I}.$$

**Discretization:** The transition distribution $p(\boldsymbol{w}(t_k)|\boldsymbol{w}(t_{k-1}))$ of the SDE is a Gaussian distribution following Lemma A.9 in Särkkä & Svensson [54]. We consider a sufficiently small interval $(t_{k-1}, t_k)$ with $\Delta t = t_k - t_{k-1} = \eta$, and assume that both the gradient $g(\boldsymbol{m},t)$ and the variance term of $L(\boldsymbol{m},t)$ remain approximately constant over this interval, denoted by $g_k$ and $\sigma_k^2$. Under the constant-gradient assumption, we also take $\mathcal{G}(\boldsymbol{m},t) = \boldsymbol{0}$. Solving the resulting ordinary differential equations yields the mean and covariance of the transition distribution:

$$\mu_{k|k-1} = \boldsymbol{m}(t_k|t_{k-1})$$
$$= \boldsymbol{w}(t_{k-1}) - \int_{t_{k-1}}^{t_{k-1}+\Delta t} g_k\, \mathrm{d}t$$
$$= \boldsymbol{w}(t_{k-1}) - g_k\Delta t,$$
$$\Sigma_{k|k-1} = \mathbf{S}(t_k|t_{k-1})$$
$$= \int_{t_{k-1}}^{t_{k-1}+\Delta t} \eta\sigma_k^2\mathbf{I}\, \mathrm{d}t$$
$$= \sigma_k^2\Delta t^2\mathbf{I},$$

where $\boldsymbol{m}(t_{k-1}) = \boldsymbol{w}(t_{k-1})$ and $\mathbf{S}(t_{k-1}) = \boldsymbol{0}$.

$\square$

## A.2 Justification of Small Variance

Our derivation assumes the gradient and variance are approximately constant during each discretization step. This is justified by the short time interval involved in transitioning from continuous to discrete time.

The time interval is determined by the learning rate $\eta$, and we adopt a learning rate in the range of $10^{-5}$ to $10^{-6}$, which is about 100 times smaller than that used during source model training (typically $10^{-3}$ to $10^{-4}$). Empirically, as shown in Figure 6 (x-axis), the maximum observed variance is around $e^{-11}$ and the minimum is approximately $e^{-28}$. This result demonstrates that variance remains extremely small on continual TTA settings. This behavior is consistent across multiple datasets, as shown in Figure 8.

This consistency can be explained by the fact that TTA assumes a well-trained model as its starting point (see Section 2.1), resulting in minimal model drift during adaptation. Consequently, the assumption of approximately constant mean and variance is naturally satisfied in the TTA setting.

### A.3  Comparison of Variance and Log-variance

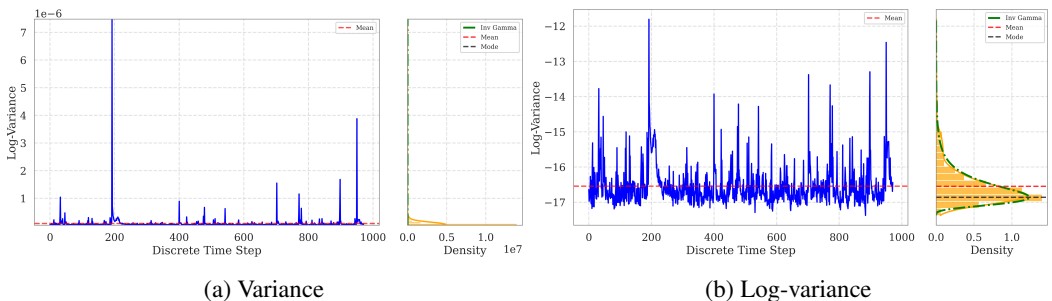

(a) Variance

(b) Log-variance

Figure 8: Variance (left) and log-variance (right) dynamics under SL settings in i.i.d. environments. The variance signal exhibits strong fluctuations and a heavy-tailed distribution, complicating statistical modeling. In contrast, the log-variance shows more stationary behavior and aligns well with the IG distribution, enabling stable estimation and interpretation.

## B  Structured Inverse-Gamma Model Alignment

### B.1  Illustration of Algorithm

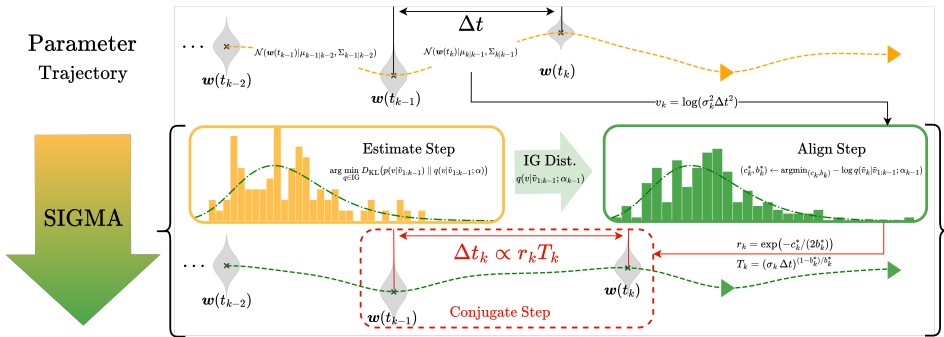

Figure 9: Illustration of the SIGMA algorithm. The black crosses represent the model parameters at each discrete time step, and the gray curves depict their corresponding distributions $p(\boldsymbol{w}(t_{1:k})) = \mathcal{N}(\boldsymbol{w}(t_{1:k})|\mu_k, \Sigma_k)$. In *Estimate Step*, SIGMA estimates an IG distribution using the log-variance values $v_{1:k-1}$ derived from the covariance of the transition distribution $p(\boldsymbol{w}(t_k)|\boldsymbol{w}(t_{k-1})) = \mathcal{N}(\boldsymbol{w}(t_k)|\mu_{k|k-1}, \Sigma_{k|k-1})$ up to time $t_{k-1}$. In *Align Step*, it computes the optimal correction terms $c_k^*$ and $b_k^*$ to align the current log-variance with the estimated IG distribution. In *Conjugate Step*, these corrections adjust the update interval $\Delta t_k$, thereby controlling the parameter dynamics at time step $t_k$.

### B.2  Derivation of Algorithm

In this section, we provide the derivation of Eq. (10). Using Eq. (7) and the log-variance, we get

$$\tilde{v}_k = \frac{v_k - c_k}{b_k} + 2\log\lambda = \frac{2\log(\sigma_k\Delta t) - c_k}{b_k} + 2\log\lambda.$$

Substituting and simplifying, we get:

$$\frac{\tilde{v}_k}{2} = \frac{\log(\sigma_k\Delta t) - \frac{1}{2}c_k}{b_k} + \log\lambda.$$

Taking the exponential of both sides:

$$\exp\left(\frac{\tilde{v}_k}{2}\right) = \lambda\exp\left(-\frac{c_k}{2b_k}\right)\exp\left(\frac{\log(\sigma_k\Delta t)}{b_k}\right) = \lambda r_k T_k \sigma_k \Delta t,$$

where $\exp\left(\log(\sigma_k\Delta t)/b_k\right) = (\sigma_k\Delta t)^{1-b_k/b_k}(\sigma_k\Delta t) = T_k(\sigma_k\Delta t)$. Taking the logarithm of both sides:

$$\tilde{v}_k = 2\log|\sigma_k(\lambda r_k T_k)\Delta t|.$$

### B.3 Pseudocode

---
**Algorithm 1** Structured Inverse-Gamma Model Alignment

---
**Require:** Initial interval $\Delta t$, Source model $f(.;\hat{w}_0)$, Alignment Strength $\lambda$,
**Initialization:** $\mu_0 = \hat{w}_0, \Sigma_0 = \mathbf{0}, \hat{g}_0 = \mathbf{0}, \tilde{v}_{1:1} = \{\}$
**for** $k = 1$ **to** $K$ **do**
    $g_k \leftarrow \nabla_{\mu_k} G(\mu_k, t_k)$                                                  $\triangleright$ Eq. (2)
    $\hat{g}_k \leftarrow (g_k + \hat{g}_{k-1})$
    $\bar{g}_k \leftarrow 1/k\ \hat{g}_k$

    $\sigma_k^2 \leftarrow 1/d\ \mathrm{Tr}[(g_k - \bar{g}_k)(g_k - \bar{g}_k)^\top]$
    $v_k \leftarrow \log(\sigma_k^2 \Delta t^2)$
    $\Delta t_k \leftarrow \Delta t$
    $(c_k^*, b_k^*) \leftarrow (0, 1)$
    **if** $|\tilde{v}_{1:k}| > 1$ **then**
        **Estimate Step:**
        $q(v|\tilde{v}_{1:k-1}; \alpha_{k-1}) \leftarrow \arg\min_{q \in \mathrm{IG}} D_{\mathbb{KL}}(p(v|\tilde{v}_{1:k-1}) \parallel q(v|\tilde{v}_{1:k-1}; \alpha))$        $\triangleright$ Eq. (8)
        **Align Step:**
        $\tilde{v}_k \leftarrow (v_k - c_k)/b_k + 2\log\lambda$
        $(c_k^*, b_k^*) \leftarrow \arg\min_{(c_k, b_k)} -\log q(\tilde{v}_k|\tilde{v}_{1:k-1}; \alpha_{k-1})$                    $\triangleright$ Eq. (9)
        $r_k \leftarrow \exp(-c_k^*/(2b_k^*)r)$
        $T_k \leftarrow (\sigma_k \Delta t)^{(1-b_k^*)/b_k^*}$
        $\Delta t_k \leftarrow (\lambda\, r_k\, T_k)\,\Delta t$
    **end if**
    **Conjugate Step:**
    $(\mu_k, \Sigma_k) \leftarrow (\mu_{k-1} - g_k\,\Delta t_k, \Sigma_{k-1} + \sigma_k^2\,\Delta t_k^2\,\mathbf{I})$             $\triangleright$ Recursion of Eq. (13)
    **if** $k > 1$ **then**
        $\tilde{v}_k \leftarrow (\log(\sigma_k^2\,\Delta t^2) - c_k^*)/b_k^* + 2\log\lambda$
        $\tilde{v}_{1:k} \leftarrow \tilde{v}_{1:k-1} \cup \{\tilde{v}_k\}$
    **end if**
**end for**

---

## C Extended Experiments

To evaluate the generality and practicality of our method, we conducted a broad set of extended experiments. These include studies across diverse model architectures, runtime efficiency, domain ordering, repeated datasets, various datasets, and the impact of alignment strength on performance.

### C.1 Diverse Model Architectures

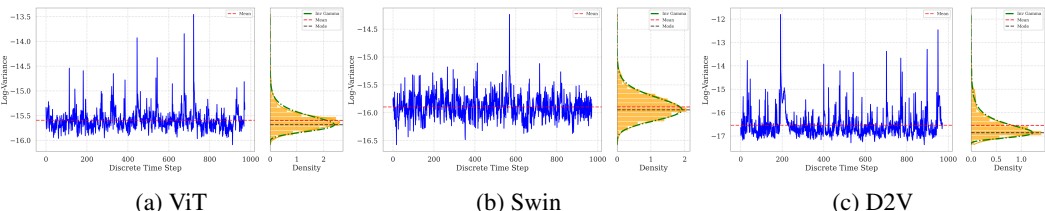

|         |          |         |
|---------|----------|---------|
| (a) ViT | (b) Swin | (c) D2V |

Figure 10: Log-variance dynamics for each backbone architecture under the stable adaptation setting (i.e., SL in i.i.d. environments) on ImageNet-C. The blue line represents the evolution of log-variance over time. Orange boxes show empirical distributions. The green dash-dot line denotes the fitted IG distribution, with red and black dashed lines representing its mean and mode, respectively.

**Justification of IG Alignment.** To verify the robustness of IG alignment assumption, we measured the log-variance under the stable adaptation setting (i.e., SL in i.i.d. environments) using three different backbone architectures: ViT, Swin, and D2V. As shown in Figure 10, the temporal evolution of log-variance (blue line) exhibited distinct patterns across models. However, in all cases, the

resulting log-variance portraits (orange boxes) showed strong GoF with the IG distribution (green dash-dot line). This consistency confirms that IG alignment is a model-agnostic property of stable adaptation dynamics.

Table 4: AERs (%) and corresponding standard deviations for ViT in Correlated Input on ImageNet-C. The **bold** number indicates the best result.

| Method | Noise | | | Blur | | | | Weather | | | | Digital | | | | AER |
|---|---|---|---|---|---|---|---|---|---|---|---|---|---|---|---|---|
| | gaussian | shot | impulse | defocus | glass | motion | zoom | snow | frost | fog | bright | contrast | elastic | pixelate | jpeg | |
| Source | 65.8 | 67.2 | 65.4 | 68.8 | 74.4 | 64.3 | 66.6 | 56.9 | 45.3 | 48.7 | 29.3 | 81.8 | 57.1 | 60.8 | 50.2 | 60.2 |
| RoTTA | 65.8 | 67.1 | 64.9 | 68.9 | 73.3 | 62.8 | 65.2 | 55.6 | 44.1 | 45.7 | 27.9 | 80.3 | 54.5 | 60.0 | 49.8 | 59.1±0.05 |
| SAR | 61.3 | 55.7 | 54.4 | 62.0 | 61.4 | 53.8 | 57.0 | 53.9 | 45.1 | 45.9 | 29.1 | 55.1 | 51.5 | 49.2 | 40.3 | 51.7±0.14 |
| TENT | 63.8 | 61.7 | 59.9 | 67.2 | 71.0 | 60.8 | 63.8 | 55.6 | 46.5 | 47.9 | 28.5 | 71.8 | 55.4 | 54.5 | 46.5 | 57.0±0.04 |
| +SIGMA | 63.8 | 61.7 | 59.3 | 63.9 | 64.2 | 53.9 | 57.7 | 53.9 | 44.6 | 40.4 | 27.4 | 59.1 | 52.9 | 47.6 | 41.0 | 52.8±0.07 |
| EATA | 61.8 | 57.1 | 56.2 | 61.5 | 62.5 | 54.4 | 57.2 | 51.8 | 45.0 | 42.5 | 28.0 | 56.8 | 51.7 | 48.4 | 42.4 | 51.8±0.10 |
| +SIGMA | 61.8 | 57.1 | 55.5 | 57.6 | 55.0 | 49.2 | 50.4 | 47.8 | 40.8 | 37.6 | 26.7 | 48.0 | 46.2 | 41.1 | 36.5 | 47.4±0.08 |
| DeYO | 60.8 | 56.4 | 55.6 | 60.5 | 61.0 | 52.4 | 57.9 | 51.7 | 42.9 | 39.1 | 27.2 | 53.1 | 51.6 | 46.8 | 41.2 | 50.5±0.11 |
| +SIGMA | 60.8 | 56.4 | 55.2 | 57.4 | 57.5 | 50.3 | 74.1 | 53.6 | 41.0 | 37.5 | 26.6 | 48.4 | 47.2 | 41.8 | 37.0 | 49.7±0.34 |
| ROID | 57.6 | 51.5 | 52.2 | 55.1 | 52.4 | 46.5 | 47.2 | 45.6 | 39.5 | 36.0 | 26.0 | 45.0 | 43.8 | 39.7 | 36.3 | 45.0±0.09 |
| +SIGMA | 57.9 | 51.7 | 52.1 | 54.4 | 50.5 | 45.3 | 44.4 | 44.3 | 39.4 | 35.5 | 26.1 | 45.3 | 39.6 | 37.5 | 35.2 | 43.9±0.13 |

Table 5: AERs (%) and corresponding standard deviations for Swin in Correlated Input on ImageNet-C. The **bold** number indicates the best result.

| Method | Noise | | | Blur | | | | Weather | | | | Digital | | | | AER |
|---|---|---|---|---|---|---|---|---|---|---|---|---|---|---|---|---|
| | gaussian | shot | impulse | defocus | glass | motion | zoom | snow | frost | fog | bright | contrast | elastic | pixelate | jpeg | |
| Source | 71.0 | 70.0 | 75.4 | 72.8 | 81.6 | 63.8 | 68.2 | 57.9 | 50.7 | 40.7 | 28.6 | 60.6 | 72.1 | 86.6 | 59.3 | 64.0 |
| RoTTA | 71.0 | 69.3 | 73.8 | 73.2 | 80.4 | 62.7 | 67.2 | 56.9 | 48.7 | 42.9 | 29.1 | 59.0 | 69.2 | 88.7 | 59.0 | 63.4±0.01 |
| SAR | 63.5 | 57.4 | 58.0 | 77.1 | 73.8 | 68.0 | 71.7 | 65.5 | 67.8 | 63.3 | 32.0 | 70.2 | 71.8 | 84.8 | 63.0 | 65.9±1.27 |
| TENT | 67.3 | 63.4 | 67.7 | 78.2 | 80.1 | 64.4 | 67.8 | 58.6 | 55.0 | 55.2 | 29.8 | 57.0 | 70.8 | 80.4 | 57.3 | 63.5±0.03 |
| +SIGMA | 67.3 | 63.4 | 66.4 | 77.9 | 75.9 | 62.7 | 64.5 | 61.6 | 55.5 | 51.2 | 26.7 | 50.4 | 69.3 | 75.9 | 51.7 | 61.4±0.25 |
| EATA | 63.0 | 56.8 | 57.6 | 68.4 | 66.8 | 54.6 | 55.7 | 52.3 | 46.7 | 42.1 | 25.9 | 48.8 | 57.1 | 64.8 | 49.7 | 54.0±0.13 |
| +SIGMA | 63.0 | 56.8 | 57.2 | 65.9 | 63.9 | 52.9 | 50.9 | 48.6 | 44.4 | 39.8 | 24.8 | 46.5 | 50.9 | 58.5 | 44.3 | 51.2±0.05 |
| DeYO | 62.6 | 56.8 | 57.3 | 72.7 | 68.9 | 58.3 | 62.4 | 52.8 | 46.7 | 76.5 | 26.7 | 48.3 | 58.2 | 64.4 | 49.6 | 57.8±1.66 |
| +SIGMA | 62.6 | 56.8 | 57.3 | 71.8 | 68.9 | 58.7 | 62.2 | 51.6 | 45.0 | 65.1 | 25.9 | 47.6 | 53.5 | 60.3 | 46.9 | 55.6±1.89 |
| ROID | 58.0 | 51.6 | 51.4 | 62.9 | 57.6 | 49.9 | 47.5 | 44.2 | 39.9 | 36.2 | 24.2 | 43.9 | 44.5 | 50.4 | 42.5 | 47.0±0.26 |
| +SIGMA | 58.7 | 52.0 | 52.0 | 64.3 | 57.6 | 49.1 | 45.8 | 42.2 | 38.7 | 34.4 | 25.1 | 43.8 | 41.9 | 46.6 | 39.8 | 46.1±0.17 |

**Benchmark Results.** Building on IG alignment principle, we applied SIGMA to both ViT and Swin backbones across multiple TTA methods. Table 4 reports results on ViT. SIGMA improved performance by 4.2% with TENT, 4.4% with EATA, 0.8% with DeYO, and 1.1% with ROID. Similar trends were observed in Table 5 for Swin, with gains of 2.1% (TENT), 3.8% (EATA), 2.2% (DeYO), and 0.9% (ROID), respectively. These consistent improvements reinforce the generality of IG alignment principle and its effectiveness across different model architectures.

## C.2 Runtime Efficiency

Efficiency is a critical factor in online adaptation scenarios. To quantify runtime overhead, we use GPU wall time (in milliseconds per sample) as our efficiency metric. Figure 11 illustrates the wall time and RAER before and after applying SIGMA for each sample-filtering method under the Correlated Input scenario on ImageNet-C. SIGMA introduced minimal additional overhead (e.g., only 0.04 ms / sample for EATA and ROID) while achieving substantial RAER improvements of 10.4% and 4.8%, respectively. These results demonstrate the high efficiency of SIGMA, which stems from its lightweight design based on scalar variance observations and derivative-free optimization. Overall, SIGMA maintains a strong balance between computational cost and adaptation effectiveness.

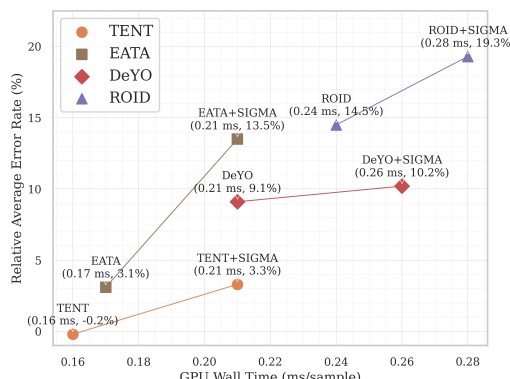

Figure 11: RAERs and corresponding GPU wall time for each method w/o and w/ SIGMA on ImageNet-C.

## C.3 Various Domain Ordering

In real-world online adaptation scenarios, the sequence in which domains appear can vary significantly, and robustness to such variation is essential for practical deployment. To evaluate this, we tested different permutations of the four broader domain categories in ImageNet-C: Noise (N), Blur (B), Weather (W), and Digital (D). Specifically, we evaluated four domain orderings: the original order N $\rightarrow$ B $\rightarrow$ W $\rightarrow$ D, along with three alternative sequences (i.e., B $\rightarrow$ W $\rightarrow$ D $\rightarrow$ N, W $\rightarrow$ D $\rightarrow$ N $\rightarrow$ B, and D $\rightarrow$ N $\rightarrow$ B $\rightarrow$ W). Figure 12 presents the AERs for each TTA method with and without SIGMA across these sequences. In all cases, SIGMA consistently improved performance, regardless of domain order. These results underscore SIGMA's robustness to domain sequence variation and confirm its practical utility in dynamic and unpredictable real-world environments.

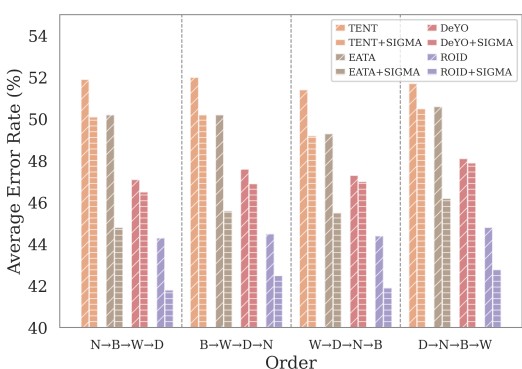

Figure 12: AERs in different domain orders for each method w/o and w/ SIGMA on ImageNet-C.

## C.4 Impact of Alignment Strength on Performance

To examine the impact of alignment strength, we focused on ROID and systematically varied the value of the hyperparameter $\lambda$ by introducing scaled perturbations. We began with a base value of $\lambda = 7.5 \times 10^{-5}$ and added an increment of $n\epsilon$, where $\epsilon = 10^{-8}$ and $n$ increased multiplicatively by a factor of 2. Figure 13 presents the AERs and corresponding standard deviations for each $\lambda$ setting. We observed that increasing alignment strength from $n = 2$ to $n = 8192$ gradually reduced the AER from approximately 41.7% to 40.8%, indicating improved performance. However, when $n$ reached 16384, the AER sharply increased to 53.6%, demonstrating that excessive alignment strength impaired performance. These results confirmed that moderate increases in alignment strength enhanced adaptation, while overly aggressive regularization degraded it. Consequently, SIGMA maintains robust performance across a broad range of $\lambda$ values as long as overly aggressive regularization is avoided.

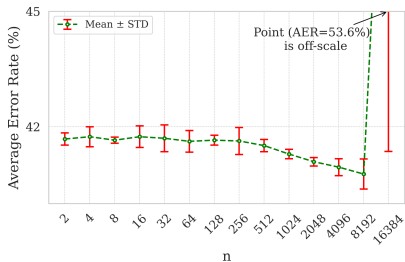

Figure 13: Effect of additional alignment strength on AER. Green circle markers denote mean AER; red bars indicate standard deviation.

## C.5 Repeated Datasets

To assess practical applicability in real-world usage patterns, we evaluated a repeated dataset scenario in which the same sequence of domains was presented to the model multiple times. This setting mimics recurring distribution patterns commonly observed in daily tasks or seasonal environmental cycles. We compared the performance of ROID with and without SIGMA under repeated domain exposures. As shown in Figure 14, ROID maintained stable adaptation performance across repetitions. Notably, SIGMA consistently outperformed ROID throughout the entire sequence, keeping the AER below 41.5% across all 15 repetitions. These results suggest that SIGMA not only prevents performance degradation during initial adaptation but also sustains robust performance in long-term, cyclical deployments, demonstrating strong potential for real-world online adaptation scenarios.

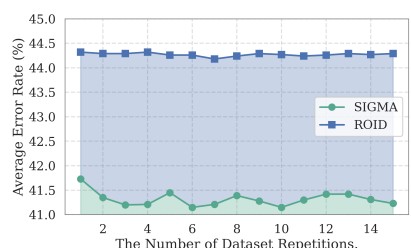

Figure 14: AERs over repeated ImageNet-C. The maximum number of repetitions is 15.

## C.6 Various Datasets

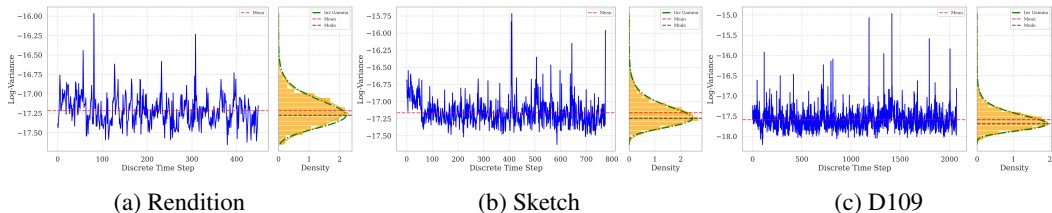

|          |          |          |
| :------: | :------: | :------: |
| (a) Rendition | (b) Sketch | (c) D109 |

Figure 15: Log-variance dynamics under the stable adaptation setting (i.e., SL in i.i.d. environments) on Rendition, Sketch, and D109. The blue line represents the evolution of log-variance over time. Orange boxes show empirical distributions. The green dash-dot line denotes the fitted IG distribution, with red and black dashed lines representing its mean and mode, respectively.

**Justification of IG alignment.** To validate the generality of IG alignment assumption, we measured the log-variance distributions under the stable adaptation setting (i.e., SL in i.i.d. environments) across diverse datasets, including Rendition, Sketch, and D109. These datasets span both single-domain (i.e., Rendition and Sketch) and multi-domain (i.e., D109) scenarios.

As shown in Figure 15, the empirical distributions of log-variance (orange boxes) across all three datasets exhibited strong alignment with the IG distribution (green dash-dot line). These findings confirms that IG alignment assumption holds across different domains and types of distribution shift.

Table 6: AERs (%) and corresponding standard deviations on Rendition and Sketch.

| Method | Rendition | Sketch |
| :--- | :---: | :---: |
| Source | 46.6 | 60.4 |
| RoTTA | 46.5±0.01 | 60.1±0.03 |
| SAR | 45.9±0.05 | 60.2±0.07 |
| TENT | 46.0±0.03 | 60.3±0.06 |
| EATA | 45.8±0.09 | 58.6±0.08 |
| DeYo | 42.9±0.07 | 60.4±0.62 |
| ROID | 41.4±0.08 | 55.7±0.02 |
| SIGMA | **37.9±0.09** | **51.5±0.12** |

**Single-Domain Settings.** We evaluated SIGMA on the Rendition and Sketch datasets by applying it to ROID, which served as a strong baseline. As reported in Table 6, SIGMA significantly improved AERs compared to ROID, achieving the best results on both datasets. These findings demonstrate that SIGMA remains effective even in specialized single-domain adaptation scenarios, further reinforcing the robustness of IG alignment principle.

Table 7: AERs (%) and corresponding standard deviations in Correlated Input on D109.

| Method | clipart | infograph | painting | real | sketch | AER |
| :--- | :---: | :---: | :---: | :---: | :---: | :---: |
| Source | 48.7 | 72.9 | 41.2 | 20.5 | 56.7 | 48.0 |
| RoTTA | 48.6 | 72.6 | 40.7 | 19.9 | 53.9 | 47.2±0.01 |
| SAR | 48.3 | 74.3 | 42.9 | 20.3 | 56.5 | 48.5±0.10 |
| TENT | 49.1 | 78.8 | 56.4 | 40.3 | 89.5 | 62.8±0.20 |
| EATA | 47.9 | 71.6 | 39.9 | 19.7 | 54.1 | 46.6±0.04 |
| DeYO | 47.2 | 74.4 | 40.9 | 19.7 | 51.2 | 46.7±0.13 |
| ROID | 43.4 | 68.5 | 37.6 | 19.3 | 50.4 | 43.8±0.03 |
| SIGMA | **42.9** | **64.8** | **36.2** | **18.3** | **44.6** | **41.4±0.03** |

Table 8: AERs (%) and corresponding standard deviations in Correlated Label on D109.

| Method | clipart | infograph | painting | real | sketch | AER |
| :--- | :---: | :---: | :---: | :---: | :---: | :---: |
| Source | 48.7 | 72.9 | 41.2 | 20.5 | 56.7 | 48.0 |
| RoTTA | 48.7 | 72.7 | 40.9 | 20.2 | 55.4 | 47.6±0.03 |
| SAR | 48.4 | 74.6 | 43.5 | 20.3 | 56.4 | 48.6±0.02 |
| TENT | 49.1 | 77.4 | 51.3 | 31.7 | 79.7 | 57.8±0.06 |
| EATA | 47.8 | 71.5 | 39.9 | 19.8 | 53.7 | 46.5±0.06 |
| DeYO | 47.3 | 74.4 | 40.6 | 19.7 | 51.0 | 46.6±0.40 |
| ROID | 43.4 | 68.0 | 37.7 | 19.4 | 50.5 | 43.8±0.06 |
| SIGMA | **43.1** | **65.0** | **36.5** | **18.9** | **45.5** | **41.8±0.06** |

**Multi-Domain Settings.** To evaluate SIGMA under natural multi-domain shifts, we used D109, a dataset that features stylistic variation across domains rather than synthetic corruptions. In Table 7 (Correlated Input) and Table 8 (Correlated Labels, $\gamma = 0.1$), SIGMA consistently outperformed ROID, confirming its effectiveness in realistic domain shift conditions. These results show that SIGMA is robust to natural distribution changes and validate the broad applicability of the IG alignment assumption beyond synthetic corruption-based benchmarks.

Table 9: AERs (%) and corresponding standard deviations on CIFAR10-C, CIFAR100-C and ImageNet-C.

| Dataset | Source | CoTTA | PETAL (FIM) | RMT | ROID | **SIGMA** |
| :--- | :---: | :---: | :---: | :---: | :---: | :---: |
| CIFAR10-C | 43.5 | 16.5±0.16 | 16.0±0.03 | 17.0±0.34 | 16.3±0.17 | **15.7±0.06** |
| CIFAR100-C | 46.4 | 32.8±0.07 | 31.3±0.13 | 30.6±0.11 | 31.7±0.11 | **29.6±0.07** |
| ImageNet-C | 64.0 | 59.3±1.23 | 58.3±0.14 | 52.6±1.00 | 47.0±0.26 | **46.1±0.17** |

**Small Multi-Domain Settings.** We conducted experiments on CIFAR10-C and CIFAR100-C for ResNet-based models [21]. For PETAL [7], we adopted the FIM variant. Table 9 presents a comparison between student-teacher methods and SIGMA applied on top of the strongest entropy-based baseline, ROID. These results show that ROID outperforms student-teacher approaches on ImageNet-C, which involves a larger number of classes. In contrast, on CIFAR10-C and CIFAR100-C, student-teacher methods such as PETAL and RMT [11] exhibit error rates lower than ROID. Specifically, PETAL achieves better performance than ROID on both CIFAR datasets, and RMT outperforms ROID on CIFAR100-C. However, SIGMA consistently improves ROID performance in all datasets, achieving error rates lower than those of all student-teacher methods, including CoTTA, PETAL, and RMT. These results demonstrate that SIGMA not only offers computational efficiency but also delivers superior accuracy compared to student-teacher models.

## C.7 Real-world Scenario

We evaluated several TTA methods in a real-time speech recognition scenario using the TEDLIUM3 dataset [23] containing streamed TED talk recordings. The validation and test sets included speech from 8 and 11 speakers, each covering different topics. Our experiments used a speech-adapted version of the D2V model pre-trained on LibriSpeech [47] as the backbone. Following the experimental protocol of SUTA [36], an established TTA method for speech recognition, we simulated realistic speaker adaptation by measuring average word error rate (WER) as new speakers sequentially entered the

Table 10: Average word error rates (WER, %) and standard deviation on validation (VALID) and test (TEST) sets in the real-world speech recognition scenario on TEDLIUM3.

| Method | VALID | TEST |
|---|---|---|
| Source | 13.0 | 12.4 |
| Pseudo Label | 12.7 ± 0.01 | 12.1 ± 0.03 |
| TENT | 12.5 ± 0.01 | 11.9 ± 0.02 |
| SUTA | 12.4 ± 0.01 | 11.6 ± 0.01 |
| **SUTA+SIGMA** | **12.3 ± 0.01** | **11.4 ± 0.01** |

stream. As shown in Table 10, SUTA+SIGMA consistently outperformed all baselines, achieving the lowest WER on both the validation and test sets. These results demonstrate that SIGMA enhances real-time adaptation by effectively mitigating speaker shift, confirming its practical applicability to real-world online adaptation tasks such as speech recognition.

# D Related Works

## D.1 Learning Dynamics

From the perspective of learning dynamics, a growing body of work demonstrates that the noise introduced by SGD exhibits heavy-tailed behavior [58, 59, 20, 57]. These studies establish that the tail behavior of SGD noise is closely related to the flatness of loss minima and, by extension, generalization performance [59, 20]. This strong theoretical understanding underpins the empirical observations reported in Appendix A.3 and substantiates our approach of treating variance as a random variable rather than a fixed value.

In parallel to these findings regarding learning dynamics and noise behavior, recent work has reinterpreted SGD-like updates from a Bayesian perspective. For instance, the existing works [41, 8] model SGD as an SDE where the stationary distribution approximates a Gaussian posterior. This work offers a principled explanation for the implicit regularization effects of SGD. Based on this view, Maddox et al. [38] have proposed posterior approximations using the trajectory of SGD itself. This work fits a Gaussian to the empirical mean and covariance of SGD iterates to enable practical Bayesian ensembling. These studies mainly focus on training from scratch or in supervised learning settings.

Distinct from prior work, we decisively extend the Bayesian viewpoint to the demanding setting of sequential domain shifts and unsupervised online adaptation. Our empirical evidence establishes that the log-variance of parameter transitions consistently aligns with the IG distribution under stable conditions. Motivated by this observation, we propose SIGMA, an algorithm that leverages this maintained alignment to deliver both stability and flexibility during online adaptation, setting it apart from existing approaches.

## D.2  Bayesian Deep Learning

DNNs are highly flexible, allowing them to represent many functions with varying levels of generalization. Using this implicit capacity is key to improving adaptability to shifts in data distributions. This concept is often applied in Bayesian deep learning, which treats model parameters as samples from an underlying distribution [49, 63, 64, 26]. Usually, the parameter distribution is approximated as a Gaussian centered around the parameter mode [8]. With this assumption, repeated training produces multiple models, which are then aggregated. Model averaging, based on this approach, has shown strong robustness to distribution shifts [25, 15, 39, 67, 65, 52].

Recent studies have extended the Bayesian perspective to the TTA setting by analyzing changes in parameter distributions during online adaptation [31, 32, 55, 30]. For example, Lee & Chang [31, 32] adopts a continual learning viewpoint, applying Bayesian filtering while explicitly fixing the transition distribution to balance information from past and present tasks. Other work [55] designs state-space models that are fundamental to Bayesian filtering and directly learn the transition distribution from the data. More recently, Lee [30] proposed modeling the transition distribution via an SDE, incorporating Bayesian filtering where the posterior is forced to converge toward a fixed value for stable adaptation. However, such approaches often overlook the natural dynamics of model parameters during real-world online adaptation. Moreover, this work forced convergence excessively suppresses heavy-tailed behavior, which degrades adaptability [58, 59, 20, 57].

In contrast, our work directly treats the variance of the transition distribution as a random variable and explicitly analyzes its temporal evolution during online learning. As a result, we uncover a key empirical law: the log-variance of parameter transitions aligns well with an IG distribution under stable online conditions. Building on this observation, we introduce a principled algorithm that estimates the IG distribution in real time and modulates the parameter dynamics accordingly. Consequently, by modeling variance as a stochastic quantity, our method avoids oversuppressing heavy-tailed behavior, thereby preserving adaptability. Furthermore, unlike conventional Bayesian filtering approaches, our method introduces no latent variables, making it lightweight and memory efficient.

# E  Limitation and Future Work

One limitation of our approach is the assumption that the parameter distribution follows a Gaussian form. While this assumption enables analytical tractability, it may limit the expressiveness of the underlying distribution. Nevertheless, our empirical results demonstrate that SIGMA, built upon this approximation, consistently performs well across various realistic scenarios and model architectures. We aim to extend our probabilistic framework for future work to account for more complex, non-Gaussian parameter distributions. We anticipate that such generalizations will be necessary in applications where greater distributional flexibility is critical. This direction may lead to more general and consequential online adaptation techniques.

