# OpenReview forum: "Parameter Dynamics of Online Machine Learning and Test-time Adaptation"
_NeurIPS.cc/2025/Conference — NeurIPS 2025 poster_

### Official Review · Reviewer_Popp · 2025-06-25

**Clarity:** 2
**Significance:** 2
**Originality:** 2
**Rating:** 3
**Confidence:** 3

**Summary:**

The paper models the dynamics of weights during online learning and test-time adaptation as a stochastic differential equation. It uses a discretization to obtain the transition distribution of the parameter evolution. The paper focuses on the transition variance and argues that the log-variance of the transition distribution follows an inverse gamma distribution when adaptation is stable. Based on this argument, the paper proposes an unsupervised adaptation mechanism that aligns the log-variance to an inverse gamma distribution to enable stable adaptation to a non-stationary test stream.

**Questions:**

- Equation 13 suggests a Bayesian treatment of the parameters. This makes sense, given that the paper focuses on the parameters' variance. However, from the text, it seems that the method is just evaluated at the mean. Could we see results with Monte Carlo sampling as well, incorporating the adjusted variance term?

- How come the adaptation gains of most TTA baselines in the standard setting of Table 1 are close to zero (e.g., for TENT, SAR, RoTTA, EATA)? These results seem to deviate from other papers \[1, 2], where the adaptation gains on ImageNet-C severity 5 are quite pronounced.

[1] Niu, Shuaicheng, et al. "Efficient test-time model adaptation without forgetting." International Conference on Machine Learning. PMLR, 2022.

[2] Zhao, Hao, et al. "On pitfalls of test-time adaptation." ICML (2023).

**Ethical Concerns:**

["NO or VERY MINOR ethics concerns only"]

**Final Justification:**

As stated in the response to the rebuttal, I think the paper needs major changes to (a) incorporate related work more thoroughly and (b) provide a better theoretical justification/explanation on why the core part of the method (IG alignment) works so well. In my opinion this should be a core part of the paper, but is currently missing entirely. I rate both of these concern as severe points and therefore believe the paper does not pass the bar for acceptance.

**Limitations:**

Limitations are not discussed.

**Quality:**

3

**Strengths And Weaknesses:**

# Strengths

- The continuous-time SDE formulation of SGD updates in online settings, while not novel [3], seems an interesting and suitable theoretical framework for analyzing dynamic adaptation methods.
- The paper is clearly written and easy to follow.

# Weaknesses

- Most related work is not (sufficiently) discussed. The following papers also model the dynamics of weights in TTA:
1. Lee, Jae-Hong, and Joon-Hyuk Chang. "Stationary latent weight inference for unreliable observations from online test-time adaptation." ICML 2024.

2. Lee, Jae-Hong, and Joon-Hyuk Chang. "Continual momentum filtering on parameter space for online test-time adaptation." ICLR 2024.

3. Lee, Jae-Hong. “Bayesian Weight Enhancement with Steady-State Adaptation for Test-time Adaptation in Dynamic Environments,” ICML 2025.

In particular, 3.) also uses an SDE formulation for parameter dynamics and adjusts the step size $\Delta t$ based on the covariance. Could we see a detailed qualitative and empirical comparison against 3.)? As the paper is very similar both conceptually as well as in the experiments to 3.), yet not cited, could the authors please highlight how the current work differs from or extends 3.)?

- The authors empirically find that the log-variance follows an inverse gamma (IG) distribution in stable adaptation settings and, in a second step, regularize the updates to follow an IG distribution. However, except for empirical correlation, the paper does not explain why the IG is the “holy grail.” What properties of the distribution explain this correlation? What is the intuition or theoretical connection behind this finding?

- If I understand correctly, adapting the model results in modifying the time discretization $\Delta t$. Does this correspond to changing the learning rate? If so, does this mean only the step size is changed but not necessarily the direction of the gradient?

---

> ### Author Rebuttal · Authors · 2025-07-30
>
> We sincerely thank the reviewer for the detailed feedback and insightful comments. We will revise the manuscript accordingly to reflect your valuable suggestions. Below, we provide our point-by-point responses to your questions.
>
> ---
>
> **W-Q1. Could the authors please highlight how the current work differs from or extends 3 ?**
>
> Thank you for recommending recent relevant studies. Below we provide a brief explanation and will include a broader and more detailed discussion in the revised manuscript.
>
> Existing works based on Bayesian deep learning primarily focus on Bayesian filtering [1.),2.),3.)]. The most recent work [3.)], in particular, emphasizes tracking the evolving parameter distribution, treating the variance as a simple variable and forcing it to converge to a fixed value. However, this approach tends to suppress the heavy-tail phenomenon excessively, which may limit adaptability [a,b,c,d]. Additionally, since these methods introduce latent parameters for filtering, they incur memory overhead comparable to the number of learnable parameters.
>
> In contrast, our work treats variance as a random variable and jointly considers both the variance distribution and the parameter distribution. This enables us to explore the intrinsic properties of the parameter distribution and empirically demonstrate that performance improves when the variance follows the IG distribution (Section 4). We further propose an algorithm that estimates the IG distribution in real time and adjusts the parameter distribution accordingly (Section 5). This allows us to regulate variance without fixing it to a specific value, avoiding excessive suppression of heavy tails and achieving superior performance. Moreover, since we do not introduce latent parameters, our method is memory-efficient.
>
> We will incorporate this discussion to strengthen the Originality and Significance of our contribution.
>
> ---
>
> **W-Q2. What properties of the distribution explain this correlation? What is the intuition or theoretical connection behind this finding?**
>
> Theoretically, the IG distribution is the conjugate prior for the variance of a Gaussian likelihood. However, its heavy-tailed nature makes empirical analysis challenging (Section 3.2, Appendix E). To observe heavy-tailed behavior, we conduct analysis and alignment in log-scale (Sections 4 and 5).
>
> We will add this explanation to improve the Clarity of our presentation.
>
> ---
>
> **W-Q3. Does this correspond to changing the learning rate? If so, does this mean only the step size is changed but not necessarily the direction of the gradient?**
>
> Thank you for this insightful question. Our IG alignment strategy derives a simple mechanism for adjusting the time interval when projecting from continuous-time to discrete-time space (Section 5 and Appendix B). This can be interpreted as an automatic, environment-specific dynamic adjustment of the learning rate without recomputing the gradient.
>
> Compared to methods like SAR, which explicitly consider the gradient direction, our mechanism consistently achieves higher performance (Tables 1–5). Furthermore, this mechanism contributes to the computational efficiency of our algorithm, which is crucial for online machine learning (Appendix D.2).
>
> ---
>
> **Q1. Could we see results with Monte Carlo sampling as well, incorporating the adjusted variance term?**
>
> Thank you for raising this important point. We conducted Monte Carlo sampling over 10 iterations and present the results below for the ROID-based SIGMA experiments under the conditions in Tables 1–3:
>
> |  | Correlated Input | Correlated Label ($\gamma$ = 0.1) | Correlated Label ($\gamma$ = 0.0) |
> | --- | --- | --- | --- |
> | Plug-in Approximation in Eq. (13) | 41.8±0.14 | 37.7±0.01 | 39.8±0.05 |
> | Monte Carlo Sampling | 41.6±0.13 | 37.6±0.04 | 39.8±0.05  |
>
> While Monte Carlo sampling led to marginal improvements, the difference was minimal. This is likely because the model is already well-trained, resulting in only minor changes during adaptation. Indeed, the mean of the variance remains very small (between $e^{-21}$ and $e^{-23}$) as shown in Figure 6(d).
>
> Moreover, Monte Carlo sampling requires repeated predictions, increasing inference time by a factor of 10, which significantly impacts efficiency in online learning. This result supports our decision to use the plug-in approximation over Monte Carlo sampling.
>
> We will include this discussion to enhance the Significance of our work.
>
> ---
>
> **Q2. How come the adaptation gains of most TTA baselines in the standard setting of Table 1 are close to zero (e.g., for TENT, SAR, RoTTA, EATA)?**
>
> We strictly followed the experimental protocol of the latest open-source standard TTA benchmark (Section 6 and Appendix C). Our implementation uses self-supervised ViT models, consistent with those in recent literatures [1.),2.),3.)], and the baseline trends we observe are similar.
>
> On the other hand, [1,2] primarily use less powerful ResNet or pure ViT backbones. We have also evaluated our method on both pure ViT and Swin architectures, as reported in Tables 4 and 5, and obtained results that align closely with the findings in [1,2].
>
> **References**
>
> [a] A tail-index analysis of stochastic gradient noise in deep neural networks. ICML 2019.
>
> [b] Hausdorff dimension, heavy tails, and generalization in neural networks. NeurIPS 2020.
>
> [c] The heavy-tail phenomenon in SGD. ICML 2021.
>
> [d] On the heavy-tailed theory of stochastic gradient descent for deep neural networks. arXiv preprint 2019.

---

> > ### Comment · Reviewer_Popp · 2025-08-04
> >
> > Thanks to the authors for their response.
> >
> > Questions 1 & 2 have been answered.
> >
> > W-Q1: Thank you for the distinction between your work and [3]. I am still somewhat concerned that some of the most closely related work have not been discussed in the original manuscript.
> >
> > W-Q2: The fact that the IG distribution is the conjugate prior for the variance of a Gaussian likelihood does not explain to me why the IG is "the holy grail" for online learning. This concern remains not sufficiently addressed.
> >
> > Because of the above concerns, I think the paper needs major changes to (a) incorporate related work more thoroughly and (b) provide a better theoretical justification/explanation on why the core part of the method (IG alignment) works so well. Therefore, I will keep my score for now.

---

> ### Author Response · Authors · 2025-08-05
>
> Thank you for your response to our earlier rebuttals. We appreciate the opportunity to provide further clarification on the points you raised.
>
> ---
>
> **W-Q1+:**
>
> As you correctly noted, it would have been ideal for our original manuscript to include a discussion of related work [3]. However, at the time of submission, this work had not yet been made publicly available. As mentioned in our previous response to W-Q1, we will incorporate a detailed discussion of this work and its relationship to ours in the revised manuscript.
>
> ---
>
> **W-Q2+:**
>
> To further elaborate on our previous response to W-Q2, we would like to clarify the role of the IG distribution from a Bayesian perspective.
>
> Our theoretical background is Bayesian statistics, where prior distribution is interpreted as beliefs and learning is viewed as the process of updating those beliefs in light of observed data. Within this framework, conjugate priors are commonly adopted.
>
> In our case, we treat the IG distribution as a prior over the log-variance, representing our belief about the dynamics of the parameter distribution. This belief is then tested against real-time empirical variance observations during online adaptation (Section 4), and subsequently updated through the algorithm we propose (Section 5).
>
> To be clear, we do not claim that the IG distribution is a universal solution or "holy grail" for all online learning scenarios. Instead, our claim is more modest and empirically grounded: under stable online conditions, we observe that the log-variance of SGD-induced parameter transitions aligns well with an IG distribution. Enforcing this alignment during adaptation leads to improved performance in practice.
>
> ---
>
> We hope this clarification helps address your remaining concerns. We appreciate your thoughtful feedback and will reflect these discussions in the revised manuscript.

---

> > ### Comment · Reviewer_Popp · 2025-08-07
> >
> > Thank you for providing additional explanation. It is much appreciated.
> >
> > ### W-Q2
> > Thank you for the additional comments. A follow-up question to better understand why the IG distribution is correlated with stable adaptation: Based on your comments, it suggests that the alignment method can be (somewhat) interpreted as bringing the posterior distribution of the variance closer to the prior distribution. Is this interpretation reasonable?
> >
> > I believe that providing a theoretical interpretation (if possible) would strengthen the paper. The lack of a theoretical motivation has also been noted by Reviewer TMwY.
> >
> > ### W-Q1
> > Since Reviewer ePCh also raised concerns about missing related work, I would appreciate it if the authors could post their updated related work section.
> >
> > I am particularly concerned by the following statement in line 689:
> >
> > *“To our knowledge, no prior work has modeled the temporal evolution of parameter distributions during the adaptation process. This work aims to fill this gap by employing SDEs to model a time-evolving Gaussian distribution over model parameters.”*
> >
> > [1,2,3,4] have modeled the temporal evolution of parameter distributions during test-time adaptation. I suggest revising the stated contribution and incorporating [1–4].
> >
> > [1] Lee, Jae-Hong, and Joon-Hyuk Chang. "Stationary latent weight inference for unreliable observations from online test-time adaptation." ICML 2024.
> >
> > [2] Lee, Jae-Hong, and Joon-Hyuk Chang. "Continual momentum filtering on parameter space for online test-time adaptation." ICLR 2024.
> >
> > [3] Lee, Jae-Hong. “Bayesian Weight Enhancement with Steady-State Adaptation for Test-time Adaptation in Dynamic Environments,” ICML 2025.
> >
> > [4] Schirmer, Mona, Dan Zhang, and Eric Nalisnick. "Test-time adaptation with state-space models." ICML 2024 Workshop on Structured Probabilistic Inference & Generative Modeling.

---

> > > ### Author Response · Authors · 2025-08-08
> > >
> > > **W-Q2++**
> > >
> > > Thank you for your insightful follow-up and for raising this important point regarding the theoretical underpinnings of the IG distribution’s role in stable adaptation. Your interpretation is generally reasonable. More specifically, the IG alignment approach can be seen as iteratively estimating a posterior of the variance, which is then reused as a prior in subsequent adaptation steps. When a significant mismatch between the prior and posterior is observed, our method effectively adjusts the observations to minimize this discrepancy. Inspired by your suggestion, we plan to include this interpretation in Section 5 of the revised manuscript.
> > >
> > > Regarding the theoretical motivation for why the IG distribution is naturally aligned with stable adaptation, our insight is rooted in a principle of constrained maximum entropy. Specifically, we consider the distribution over a variance-related random variable that supports adaptability through heavy-tailed behavior (i.e., maintaining sufficient flexibility), while still achieving stability through the notion of a stationary distribution governed by fixed moment conditions.
> > >
> > > Let $p(x)$ be the distribution over $x$. To ensure the distribution supports rich uncertainty (i.e., flexibility), we want to maximize entropy $H[p]=−\int p \log p $. At the same time, to enforce stability, we impose fixed moment constraints. Concretely, we consider the following moment conditions: the first moment of $\log x$ is fixed to $m_1$, promoting heavy-tailed behavior in log-space, and the first moment of $1/x$ is fixed to $m_2$, enforcing control over the tail density near zero.
> > >
> > > This leads us to a constrained optimization problem using Lagrange multipliers $\lambda_0, \lambda_1, \lambda_2$:
> > >
> > > $\mathcal{L} (p)
> > > = -\int p\log p -\lambda_0\Bigl(\int p -1\Bigr) -\lambda_1\Bigl(\int p\log x - m_1\Bigr) -\lambda_2\Bigl(\int p\tfrac1x - m_2\Bigr),$
> > >
> > > where $\int p -1$ is the sum of probability constraints. Solving for the stationary condition $\delta \mathcal{L} / \delta p = 0$ gives:
> > >
> > > $\log p(x) = - 1 -\lambda_0 -\lambda_1\log x -\lambda_2\frac{1}{x}$
> > >
> > > which simplifies to:
> > >
> > > $p(x) = C x^{-\lambda_1} \exp\Bigl(-\lambda_2\tfrac1x\Bigr).$
> > >
> > > where $C = \exp(-1 - \lambda_0)$. By identifying parameters as $\lambda_1 = \alpha + 1,\lambda_2 = \beta$, we recover the (unnormalized) IG distribution:
> > >
> > > $p(x) = C x^{-(\alpha+1)}\exp\bigl(-\beta/x\bigr).$
> > >
> > > Upon normalizing, this yields the full IG distribution. Therefore, under fixed-moment constraints and maximum entropy, the IG distribution naturally emerges as the optimal form, balancing flexibility (via heavy tails) and stability (via fixed statistical moments). This theoretical framing helps explain why modeling variance dynamics via an IG prior leads to more robust adaptation.
> > >
> > > The finalized version will include this analysis in Appendix A to strengthen the motivation for our approach and address the concern regarding theoretical justification.

---

> ### Author Response · Authors · 2025-08-08
>
> **W-Q1++**
>
> We incorporated the discussions with Reviewer ePCh by restructuring the related work section and assigning a new subsection titled Learning Dynamics. The current related work (i.e., Bayesian Deep Learning) was moved into a separate subsection. Your suggestions have been reflected in this subsection, and we have also included an additional relevant study [4] that aligns with the works you previously recommended.
>
> The revised subsection (beginning around original line 687) is as follows:
>
> > Recent studies have extended the Bayesian perspective to the TTA setting by analyzing changes in parameter distributions during online adaptation [1, 2, 3, 4]. For example, [1, 2] adopt a continual learning viewpoint, applying Bayesian filtering while explicitly fixing the transition distribution to balance information from past and present tasks. Other works, such as [4], design state-space models that are foundational to Bayesian filtering and directly learn the transition distribution from the data. More recently, [3] proposes modeling the transition distribution via an SDE, incorporating Bayesian filtering where the posterior is forcibly constrained to converge toward a fixed value for stable adaptation. However, such approaches often overlook the natural dynamics of model parameters during real-world online adaptation. Moreover, this work forced convergence excessively suppresses heavy-tailed behavior, which degrades adaptability [a, b, c, d].
> >
> > In contrast, our work directly treats the variance of the transition distribution as a random variable and explicitly analyzes its temporal evolution during online learning. This approach allows us to uncover a key statistical insight: under stable online conditions, we observe that the log-variance of parameter transitions aligns well with an IG distribution. Building on this observation, we introduce a principled algorithm that estimates the IG distribution in real time and modulates parameter dynamics accordingly. By modeling variance as a stochastic quantity, our method avoids over-suppressing heavy-tailed behavior, thereby preserving adaptability. In addition, unlike conventional Bayesian filtering approaches, our method introduces no latent variables, making it both lightweight and memory-efficient.
> >
>
> We believe this revision clarifies our contribution and more accurately positions our work within the existing literature.
>
> **References**
>
> [1] Lee, Jae-Hong, and Joon-Hyuk Chang. "Stationary latent weight inference for unreliable observations from online test-time adaptation." ICML 2024.
>
> [2] Lee, Jae-Hong, and Joon-Hyuk Chang. "Continual momentum filtering on parameter space for online test-time adaptation." ICLR 2024.
>
> [3] Lee, Jae-Hong. “Bayesian Weight Enhancement with Steady-State Adaptation for Test-time Adaptation in Dynamic Environments,” ICML 2025.
>
> [4] Schirmer, Mona, Dan Zhang, and Eric Nalisnick. "Test-time adaptation with state-space models." ICML 2024 Workshop on Structured Probabilistic Inference & Generative Modeling.
>
> [a] A tail-index analysis of stochastic gradient noise in deep neural networks. ICML 2019.
>
> [b] Hausdorff dimension, heavy tails, and generalization in neural networks. NeurIPS 2020.
>
> [c] The heavy-tail phenomenon in SGD. ICML 2021.
>
> [d] On the heavy-tailed theory of stochastic gradient descent for deep neural networks. arXiv preprint 2019

---

### Official Review · Reviewer_ryf8 · 2025-06-29

**Clarity:** 4
**Significance:** 4
**Originality:** 4
**Rating:** 5
**Confidence:** 5

**Summary:**

This paper investigates the parameter dynamics of online machine learning and its application in test-time adaptation (TTA). The key findings include: (1) the fitness of inverse gamma (IG) and log-variance serves as a strong predictor of adaptation performance and model collapse, and (2) state-of-the-art TTA methods often implicitly promote IG alignment. Furthermore, the authors propose SIGMA, a method that dynamically estimates an appropriate IG distribution using derivative-free optimization and adjusts the parameter update trajectory to maintain alignment during online adaptation. Experimental results demonstrate the effectiveness of SIGMA.

**Questions:**

* In line 511, authors show that DUC dataset is used. However, the motivation and details of DUC is not introduced and not any reference. DomainNet-126 [3] , a subsect of Domain Net[2] , or full DomainNet is more widely used in related papers [1,4].

* Some references should be cited with published format, such as [40], please check all references.

* How to ensure $c_k \leq 0$ and $b_k > 1$ (line 244)? use Lagrange multiplier or just clip?
* How to understand alignment length $\lambda$ (in line 245)?
* It is better to show how to solve Eq.(9) and get Eq.(10). And how to get $r_k$ and $T_k$ ?

**Reference**
1. Robust Test-Time Adaptation in Dynamic Scenarios. CVPR 2023
2. Moment Matching for Multi-Source Domain Adaptation. ICCV 2019.
3. Semi-supervised Domain Adaptation via Minimax Entropy. ICCV 2019.
4. Feature alignment and uniformity for test time adaptation. CVPR 2023.

**Ethical Concerns:**

["NO or VERY MINOR ethics concerns only"]

**Final Justification:**

I have carefully reviewed the authors' rebuttal. My concerns have been addressed, and I will keep my score (5, accept).

**Limitations:**

Please see Questions.

**Paper Formatting Concerns:**

No Paper Formatting Concern

**Quality:**

4

**Strengths And Weaknesses:**

**Strengths**
* This paper studies the fundamental and important question about parameter dynamics in online machine learning and test-time adaptation.

* Interesting and novel findings about parameter dynamics.

* A plug-and-play and effective method improves the performance of TTA methods.

**Weaknesses**
I do not find any major weakness. However, I have some questions and advise, see below.

---

> ### Author Rebuttal · Authors · 2025-07-30
>
> We thank the reviewer for the detailed comments. We will revise the manuscript to incorporate your valuable suggestions (Q1 and Q2). Below are our responses to your additional questions.
>
> ---
>
> **Q3. How to ensure $c_k\leq1$ and $b_k>1$ (line 244)? use Lagrange multiplier or just clip?**
>
> Thank you for the insightful question. As you correctly noted, we solve for the optimal values within the feasible domain and apply clipping if the solution lies outside the valid region. This provides a simple and effective way to enforce the constraints. In practice, we implement this using maximum likelihood estimation with a simplex-based optimizer that handles constraints efficiently.
>
> We will incorporate this explanation into Section 5.
>
> ---
>
> **Q4. How to understand alignment length $\lambda$ (in line 245)?**
>
> The alignment strength $\lambda$ quantifies how closely the updated distribution should follow the real-time reference distribution $q$, which is assumed to follow the IG form. A value of $\lambda=0$ implies complete distrust in the reference distribution, effectively setting $\Delta t_k=0$, which freezes parameter updates. Conversely, higher values of $\lambda$ indicate greater trust in the reference distribution and thus allow larger parameter updates.
>
> We will incorporate this explanation into Section 5.
>
> ---
>
> **Q5. how to solve Eq.(9) and get Eq.(10). And how to get $r_k$ and $T_k$?**
>
> As noted in Q3, we obtain the $c_k$ and $b_k$ from Eq. (9) within the constrained domain. Equation (10) is derived from Eq. (7):
>
> $
> \tilde{v}_k = (v_k - c_k)/b_k + 2\log \lambda
> $
>
> where $v_k = \log(\sigma_k^2 \Delta t^2) = 2\log(\sigma_k \Delta t)$. Substituting and simplifying, we get:
>
> $
> \frac{\tilde v_k}{2} = \log \lambda + \frac{\log(\sigma_k \Delta t) - \tfrac{1}{2}c_k}{b_k}.
> $
>
> Taking the exponential of both sides:
>
> $
> \exp\left(\frac{\tilde v_k}{2}\right)
> = \lambda \exp\left( -\frac{c_k}{2b_k} \right)\exp\left( \frac{\log(\sigma_k \Delta t)}{b_k} \right)=\lambda r_k T_k \sigma_k \Delta t,
> $
>
> where $\exp\left( \frac{\log(\sigma_k \Delta t)}{b_k} \right) = (\sigma_k \Delta t)^{\frac{1 - b_k}{b_k}}
> (\sigma_k \Delta t) = T_k (\sigma_k \Delta t)$.
>
> Taking the logarithm of both sides:
>
> $
> \tilde v_k = 2\log |\sigma_k (\lambda r_k T_k) \Delta t|.
> $
>
> This derivation provides a complete and interpretable expression for $\hat{v}_k$, connecting it back to the dynamics of the variance term and the alignment process.
>
> We will add this derivation to the manuscript for completeness.

---

> > ### Comment · Reviewer_ryf8 · 2025-08-01
> >
> > Thank you for your response. My questions have been answered. I keep my score.

---

> > > ### Author Response · Authors · 2025-08-05
> > >
> > > Thank you for your response. We are pleased to have the opportunity to address your questions.

---

### Official Review · Reviewer_TMwY · 2025-07-02

**Clarity:** 3
**Significance:** 3
**Originality:** 3
**Rating:** 4
**Confidence:** 4

**Summary:**

The paper proposes a probabilistic framework that models the parameter adaptation for test-time adaptation (TTA) using stochastic differential equation (SDE). The authors show empirically that for stable adaptation, the log-variance distribution of the SDE's transition aligns close to an inverse-gamma (IG) distribution and that deviating from this alignment correlates strongly with model collapse.
Thus, they propose the Structured Inverse-Gamma Model Alignment (SIGMA) algorithm, based on this observation.
SIGMA can be used with existing TTA approaches, and the authors have shown experimental results on the ImageNet-C dataset, showing improved performance for the proposed approach.

**Questions:**

1. The experiments are performed only for the ImageNet-C dataset benchmark. Whether the authors have performed experiments on other benchmarks such as CIFAR10C, CIFAR100C, etc.?
2. How does the proposed approach perform in a more non-i.i.d. setting of continual TTA [1][2]? It is also noteworthy that [2] proposes a probabilistic framework, though it is different from the SDE-based probabilistic framework proposed in this paper.
3. Could you provide experimental analysis showing how well the assumptions hold when there is a drastic distribution shift?
4. Related to question 3, the continual TTA setting will be a more realistic test for this assumption, but more broadly, how well do the authors feel that the proposed approach will generalize to the continual TTA setting?

**References**
1. Wang, Qin, et al. "Continual test-time domain adaptation." Proceedings of the IEEE/CVF Conference on Computer Vision and Pattern Recognition. 2022.
2. Brahma, Dhanajit, and Piyush Rai. "A probabilistic framework for lifelong test-time adaptation." Proceedings of the IEEE/CVF Conference on Computer Vision and Pattern Recognition. 2023.

**Ethical Concerns:**

["NO or VERY MINOR ethics concerns only"]

**Final Justification:**

The authors have addressed most of my queries.

They have reported additional comparisons with other recent CTTA approaches during the discussion.

However, these comparisons should be included in the main paper, and more discussions on related works, compared state-of-the-art approaches, and benchmark datasets should be provided.

Thus, I will keep my score unchanged.

**Limitations:**

The reliance of Theorem 1 on the strong assumption that both the gradient and the variance term remain approximately constant within each discrete time-step is a limitation that needs to be discussed.

**Quality:**

2

**Strengths And Weaknesses:**

**Strengths**
* This paper focuses on the parameter dynamics using SDEs, which is different from existing works on TTA that focus on the loss function, parameter efficiency, pseudo-labeling, or sample selection.
* The empirical observation that the log-variance of the SDE's transition aligns closely with an IG distribution is an interesting empirical result.

**Weaknesses**
* The experimental evaluation is limited. The experiments are performed only for the ImageNetC dataset benchmark, whereas existing approaches report their experiments on other benchmarks such as CIFAR10C, CIFAR100C classification, and/or Cityscapes-to-ACDC semantic segmentation.
* Theorem 1 involves questionable assumptions. The paper's theoretical claim relies on Theorem 1, which is about the discretization of the SDE approximation. This theorem relies on the assumption that both the gradient and the variance term remain approximately constant within each discrete time-step (Appendix A.1, Line 494). This is a strong assumption in the context of the non-i.i.d. setting that the paper focuses on. When the model encounters an abrupt domain shift (say, moving from noise to blur), the loss landscape and, thus, the gradients can change dramatically. This paper does not show how well this assumption holds in practice.
* The paper shows empirically that IG alignment correlates with stability, but there is no theory on the reason for this correlation. Is the correlation specific to a particular dataset, ImageNetC? Results on more datasets will help in supporting the claims made in the paper.

---

> ### Author Rebuttal · Authors · 2025-07-30
>
> We sincerely thank the reviewer for the insightful and thorough feedback. We will revise the manuscript to incorporate your valuable comments. Below, we provide detailed responses to your questions.
>
> ---
>
> **W1, Q1. The experiments are performed only for the ImageNet-C dataset benchmark. Whether the authors have performed experiments on other benchmarks such as CIFAR10C, CIFAR100C, etc.?**
>
> Recent works [a, b, c, d] have primarily focused on more realistic, large-scale datasets such as ImageNet-C, which contain a higher number of classes and are more challenging. Nonetheless, we have evaluated our method on a variety of datasets, including D109, Sketch, and Rendition, as shown in Appendix D.6. Furthermore, in Appendix D.7, we extend our method to a speech recognition task in a continual multi-speaker adaptation scenario using realistic TED talk recordings.
>
> We have also conducted experiments on CIFAR10C and CIFAR100C using ResNet-based models. Below are the results of applying SIGMA to both TENT and ROID baselines:
>
> | Dataset | Source | TENT | +SIGMA | ROID | +SIGMA |
> | --- | --- | --- | --- | --- | --- |
> | CIFAR10C | 43.5 | 19.8±0.07 | 18.2±0.06 | 16.3±0.17 | 15.7±0.06 |
> | CIFAR100C | 46.4 | 32.5±0.07 | 31.2±0.20 | 31.7±0.11 | 29.6±0.07 |
>
> As shown, our method consistently improves performance on both datasets across different baselines, including the standard method TENT and the stronger baseline ROID.
>
> We will explicitly include this explanation to improve the Quality of the paper.
>
> ---
>
> **W2, Limitations. This theorem relies on the assumption that both the gradient and the variance term remain approximately constant within each discrete time-step.**
>
> **Q3. Could you provide experimental analysis showing how well the assumptions hold when there is a drastic distribution shift?**
>
> Thank you for the insightful comment. This assumption is naturally satisfied in the online adaptation setting for the following reasons.
>
> As explained in Appendix A, our derivation assumes the gradient and variance are approximately constant during each discretization step. This assumption is justified by the small time interval involved in transitioning from continuous to discrete time.
>
> The time interval is determined by the learning rate $\eta$, and we adopt a learning rate in the range of $10^{-5}$ to $10^{-6}$—which is about 100 times smaller than that used during source model training (typically $10^{-3}$ to $10^{-4}$). Empirically, as shown in Figure 6 (x-axis), the maximum observed variance is around $e^{-11}$ and the minimum is approximately $e^{-28}$, demonstrating that variance remains extremely small on continual TTA settings. This behavior is consistent across multiple datasets, as shown in Figure 15.
>
> The reason for this consistency is that TTA assumes a well-trained model as its starting point (Section 2.1), leading to minimal model drift during adaptation. Consequently, the assumption of approximately constant mean and variance is naturally satisfied in the TTA setting.
>
> We will explicitly include this explanation to improve the Quality of the paper.
>
> ---
>
> **W3. Is the correlation specific to a particular dataset, ImageNetC? Results on more datasets will help in supporting the claims made in the paper.**
>
> Figure 15 provides validation of the IG distribution assumption across various datasets. Additionally, the results shown in Figure 10, which include diverse model architectures, further confirm the robustness of the observed pattern.
>
> ---
>
> **Q2. How does the proposed approach perform in a more non-i.i.d. setting of continual TTA [1][2]?**
>
> All experiments, including the IG distribution analysis in Section 4, are conducted under a continual TTA setting—i.e., we do not reset the parameters between adaptation episodes. Moreover, Appendix D.3 includes experiments that vary the domain order to test robustness.
>
> Tables 2, 3, and 8 further evaluate performance under more challenging non-i.i.d. scenarios, such as Label Shift.
>
> We will emphasize this setup more clearly in Section 6 to improve Clarity.
>
> ---
>
> **Q4. Related to question 3, the continual TTA setting will be a more realistic test for this assumption, but more broadly, how well do the authors feel that the proposed approach will generalize to the continual TTA setting?**
>
> Thank you for the constructive question. All of our experiments were conducted under continual TTA settings. In Appendix D, we further extend our evaluation to test generalizability across diverse model architectures (Appendix D.1), realistic scenarios such as domain order variation and long streams (Appendix D.3 and D.5), and a variety of datasets, including D109, Rendition, and Sketch (Appendix D.6).
>
> To explore its applicability in real-world settings, we also apply our algorithm to a speech recognition task involving continual multi-speaker adaptation using realistic TED talk recordings (Appendix D.7).
>
> Across all these experiments, SIGMA consistently improves performance across a wide range of settings. These results support our central claim: enforcing the IG alignment in parameter dynamics helps prevent collapse and leads to robust performance in both online and continual adaptation settings.
>
> ---
>
> **References**
>
> [a] Towards stable test-time adaptation in dynamic wild world. ICLR 2023.
>
> [b] Entropy is not Enough for Test-Time Adaptation: From the Perspective of Disentangled Factors. ICLR 2024.
>
> [c] Continual momentum filtering on parameter space for online test-time adaptation. ICLR 2024.
>
> [d] Test-Time Model Adaptation with Only Forward Passes. ICML 2024.

---

> > ### Comment · Reviewer_TMwY · 2025-08-01
> > **Acknowledgment to the Authors' Response and Further Clarifications**
> >
> > Thanks to the authors for their detailed response.
> >
> > As the authors clarify, "All of our experiments were conducted under continual TTA settings."
> > So, the comparisons against the state-of-the-art approaches like CoTTA [1], PETAL [2], and RMT [3] are notably absent.
> >
> > I will keep my score unchanged for now.
> >
> > **References**
> > 1. Wang, Qin, et al. "Continual test-time domain adaptation." Proceedings of the IEEE/CVF Conference on Computer Vision and Pattern Recognition. 2022.
> > 2. Brahma, Dhanajit, and Piyush Rai. "A probabilistic framework for lifelong test-time adaptation." Proceedings of the IEEE/CVF Conference on Computer Vision and Pattern Recognition. 2023.
> > 3. Döbler, Mario, Robert A. Marsden, and Bin Yang. "Robust mean teacher for continual and gradual test-time adaptation." Proceedings of the IEEE/CVF Conference on Computer Vision and Pattern Recognition. 2023.

---

> > > ### Author Response · Authors · 2025-08-02
> > >
> > > We sincerely thank the reviewer for the constructive feedback.
> > >
> > > Our work focuses on efficient entropy minimization-based methods. To address your concern regarding student-teacher approaches, we conducted additional comparisons with the suggested methods: CoTTA [1], PETAL [2], and RMT [3]. We will include both the results and references in the revised manuscript.
> > >
> > > For PETAL, we adopted the FIM variant, which consistently shows lower error rates in the original paper. The table below presents a comparison between the suggested methods and our method, SIGMA, applied on top of the strongest entropy-based baseline (i.e., ROID):
> > >
> > > | Dataset | Source | CoTTA | PETAL (FIM) | RMT | ROID | +SIGMA |
> > > | --- | --- | --- | --- | --- | --- | --- |
> > > | CIFAR10C | 43.5 | 16.5±0.16 | 16.0±0.03 | 17.0±0.34 | 16.3±0.17 | **15.7±0.06** |
> > > | CIFAR100C | 46.4 | 32.8±0.07 | 31.3±0.13 | 30.6±0.11 | 31.7±0.11 | **29.6±0.07** |
> > > | ImageNetC | 64.0 | 59.3±1.23 | 58.3±0.14 | 52.6±1.00 | 47.0±0.26 | **46.1±0.17** |
> > >
> > > These results show that ROID outperforms student-teacher approaches on ImageNetC, which involves a larger number of classes. In contrast, on CIFAR10C and CIFAR100C, student-teacher methods such as PETAL and RMT exhibit lower error rates than ROID. Specifically, PETAL achieves better performance than ROID on both CIFAR datasets, and RMT outperforms ROID on CIFAR100C.
> > >
> > > However, SIGMA consistently improves upon ROID across all datasets, achieving lower error rates than all student-teacher methods, including CoTTA, PETAL, and RMT. These results demonstrate that SIGMA not only offers computational efficiency but also delivers superior accuracy compared to student-teacher models.
> > >
> > > We appreciate the suggestion and believe these additions significantly strengthen the empirical contributions of the paper.

---

### Official Review · Reviewer_ePCh · 2025-07-03

**Clarity:** 3
**Significance:** 3
**Originality:** 3
**Rating:** 4
**Confidence:** 4

**Summary:**

The authors empirically find that during successful test-time adaptation (TTA) the variance of the gradients seem to follow a log-normal distribution (or rather that the log-normal distribution can fit the gradient variance well). They show in various experiments that departure from this empirical phenomenon correlates strongly with bad adaptation performance. Based on this insight, they then design an adaptive update rule that can be combined with other TTA methods in the literature to increase the adaptation performance further. The adaptive rule can scale the learning rate accordingly to improve adaptation stability. It is shown in various experiments that TTA performance improves when this adaptive rule is combined with existing state of the art TTA methods.

**Questions:**

- The paper does not concern foundation models in particular, so the mention of foundation models at the start of the abstract is misleading and inappropriate. Moreover, The refs in line 22 have nothing to do with foundation models, they are CNNs for image recognition, of course they can be used with/inside foundation models but foundation models typically mean (sometimes multi-modal) LLMs.
- Model collapse is not defined in the introduction adequately I think. Performance degradation does mean model collapse (line 157), which rather means catastrophic forgetting of the source I believe.
- The first paragraph seems to lump two different (but related) topics: domain adaptation (DA) and continual learning (CL). The references seem to be mainly from DA whereas the discussion mostly relates to CL (online ML better translates to CL, especially if the particular setting of unsupervised DA is not mentioned). The two fields have different emphasis: very roughly, CL focuses on catastrophic forgetting, and memory related issues while DA focuses on alignment of distributions to transfer well to unlabeled target distribution. Test-time UDA does focus also on the harder problem of how to align the distributions when the source inputs are not available. From the paragraph's discussion the 'collapse' seems to relate more to the 'catastrophic forgetting' studied in CL, rather than to the 'negative transfer' of UDA methods. Hence I would recommend clarifying the discussion or at least choosing the references more carefully.
- The next paragraphs seem to clarify somewhat the above distinctions, however the confusion in the first paragraph can be alleviated by explicitly mentioning/separating the UDA/CL fields in 1st paragraph explicitly.
- Section 2.1 is repetitive: I would recommend streamlining with section 1, as the content is discussed there already.
- But otherwise, the paper would definitely benefit from a wider and more detailed related works section, some of which should be streamlined with the main text. For instance, there is a whole series of research on the 'heavy tail' phenomenon in deep learning theory and practice (e.g. SGD, Hessians, learned weights etc.), which is not mentioned at all. Were the authors not influenced by this research direction? Also, Bayesian deep learning and the parameter distributions that evolve during SGD-like updates are not mentioned adequately (except briefly in the appendix).
- Theorem 1 does not seem new at all, moreover (6) is I believe incorrect: Sigma_{k|k-1} should have Sigma_{k-1} added to sigma_k^{2}\deltat^{2}I. More importantly I believe, Theorem 1 could have been derived just from the discrete case by propagating the Gaussian parameters along the gradient descent step, so it is not clear why the authors present a continuous time formulation. The parametric distributions can be recovered using, e.g., Bayesian or approximate Bayes methods. For instance in variational Bayes approaches, a lot of works present optimizers that include a KL-divergence term as a regularizer (to prevent distributional collapse), and can then formulate various forms of parametric distribution updates.
- It is not clear why we need the conjugate step in the algorithm presented in section 5.1. The align step already scales the learning rate
  and hence changes the model f_k right? Why do you need the conjugate step? Do you have ablations that show that this step is needed?
- In the experiments, it would help to explain the Correlated Label setting more clearly.
- Would be good to vary the fixed sequence of domains in the experiments, has this been done? What is the sequence used to generate the data shown in Tables 1-3?
- What is the model used in section 6? Add more details for the experiments and streamline such discussion with the main text.

Minor comments:
- In eq. (1) w_hat's dependence on t_{k-1} is confusing: in (2) we minimize that function w.r.t w_hat so t_{k-1} seems to be erroneous.

**Ethical Concerns:**

["NO or VERY MINOR ethics concerns only"]

**Final Justification:**

I am copying my response to the AC's question above:

1) Question of theoretical contribution: The authors do not claim to make a theoretical contribution, they clearly state throughout that the paper is based on an (interesting) empirical observation (of the relation between TTA performance stability and the inverse gamma distribution of the log variance). The Theorem 1 is there as a modeling framework to elucidate "the unique relationship between the variance and time interval that emerges when discretizing a continuous-time stochastic differential equation" (quoted from their rebuttal). I believe such results are common place in SGD literature (I would double check the references in the rebuttal [a]-[g] to be sure, if I had time). Perhaps the authors should tone down their statement of Theorem 1 to satisfy the AC.

2) Acceptance of paper after extensive rebuttal: I believe the paper should be accepted in any case. I thought the experiments were extensive and based on my limited experience, I don't think that ImageNet-C is very narrow as an experimental setup. Note that (as stated also in the rebuttal by the authors) the paper includes many more experiments besides the ImageNet-C setup.

**Limitations:**

I think the authors could say more about the limitations of the paper, as of now only the Gaussian assumption of the parameter distribution as the limiting factor is mentioned in the appendix.

**Quality:**

3

**Strengths And Weaknesses:**

I liked the experimental nature of the paper and the results seem quite promising. However, as mentioned below, the paper would benefit from a more thorough related works section and also on the motivation of the heavy tail phenomenon. Also, the continuous time formulation presented in Theorem 1 does not seem to be particularly useful or relevant to the rest of the paper, which is experimental and not theoretical. See below for more detailed comments on these issues.

---

> ### Author Rebuttal · Authors · 2025-07-30
>
> We sincerely thank the reviewer for the concrete and constructive feedback. We will revise the manuscript to incorporate your valuable suggestions, including Q1–Q5, Q9, and the Minor comments. Below are our detailed responses to your questions.
>
> ---
>
> **W, Q6. For instance, there is a whole series of research on the 'heavy tail' phenomenon in deep learning theory … Were the authors not influenced by this research direction?**
>
> Thank you for highlighting this important line of research. We provide a brief response here and will incorporate a more comprehensive discussion in the revised manuscript.
>
> A growing body of work has demonstrated that the noise introduced by SGD exhibits heavy-tailed behavior [a, b, c, d]. These studies suggest that the tail behavior of SGD noise is closely related to the flatness of loss minima and, by extension, generalization performance [b, c]. This theoretical understanding provides a foundation for the empirical observations reported in Appendix B.2 and supports our treatment of variance not as a fixed value but as a random variable.
>
> In parallel, recent work has reinterpreted SGD-like updates from a Bayesian perspective. For instance, [e, f] model SGD as an SDE where the stationary distribution approximates a Gaussian posterior. This work offers a principled explanation for the implicit regularization effects of SGD. Building on this view, [g] have proposed posterior approximations using the trajectory of SGD itself. This work fits a Gaussian to the empirical mean and covariance of SGD iterates to enable practical Bayesian ensembling. These studies primarily focus on training from scratch or within supervised learning settings.
>
> In contrast, our work extends this Bayesian viewpoint to the context of sequential domain shifts and unsupervised online adaptation, where the behavior of SGD dynamics is markedly different. Based on empirical evidence, we observe that the log-variance of parameter transitions aligns well with the IG distribution under stable conditions. Motivated by this observation, we propose SIGMA, an algorithm that maintains this alignment to achieve stable and flexible online adaptation.
>
> We will include this discussion to strengthen the Significance of our contribution.
>
> ---
> **W, Q7. … why the authors present a continuous time formulation. The parametric distributions can be recovered using, e.g., Bayesian or approximate Bayes methods.**
>
> Thank you for your detailed observation. We address the two points you raised in turn, as outlined below.
>
> *Regarding Eq. (6)*: In general, $\Sigma_{k|k-1}$ include the intermediate prior covariance $\Sigma_{k-1}$. However, as stated in Lemma A.9 in Appendix A.1, we explicitly initialize the covariance as $\boldsymbol{0}$. This assumption simplifies the analysis and leads to a more tractable update rule. We will make this modeling choice more explicit in the revised manuscript to improve clarity.
>
> *On approximate Bayesian methods and the motivation for the continuous-time formulation*: Beyond our use of the continuous-time formulation, it is possible to estimate the parameter distribution via approximate Bayesian methods. In particular, variational inference offers a principled approach to approximating the posterior using a chosen functional form, such as a Gaussian.
>
> However, our focus is not on modeling the full form of the parameter distribution, but rather on the unique relationship between the variance and time interval that emerges when discretizing a continuous-time stochastic differential equation. Specifically, we leverage the form $\sigma_k^2 \Delta t^2 \mathbf{I}$, which naturally arises in our framework. This approach allows us to capture the evolution of the parameter distribution and regulate it through variance dynamics. Instead of fitting the parameter distribution directly, we concentrate on aligning the variance distribution with the IG form to guide adaptation.
>
> While selecting an appropriate proposal distribution for variational inference can be a viable direction for regularization, such approaches do not directly reveal the underlying dynamics of variance in online adaptation. Investigating alternative approximate Bayesian methods in this context is an interesting direction, and we leave it as a potential area for future work.
>
> ---
>
> **Q8. The align step already scales the learning rate and hence changes the model f_k right? Why do you need the conjugate step? Do you have ablations that show that this step is needed?**
>
> In continuation of our response to W, Q7: the conjugate step provides a mathematical foundation that aligns the transition distribution derived in Eq. (1) with the predictive distribution $p(y | \boldsymbol{x}, \boldsymbol{w})$.
>
> The align step modifies only the parameter dynamics via the transition distribution. After accumulating this to form the parameter distribution, the conjugate step ensures consistency with the predictive distribution through Bayes' rule and a plug-in approximation, as shown in Eq. (13). This allows the predictive model to integrate the Gaussian mean as $f(\boldsymbol{x}_k;\boldsymbol{w})=f(\boldsymbol{x};\mu_k)$.
>
> As you correctly intuited, the algorithm could function based solely on the align step, but the conjugate step provides the theoretical justification needed to apply this in a probabilistic optimization framework.
>
> ---
>
> **Q10.  Would be good to vary the fixed sequence of domains in the experiments, has this been done? What is the sequence used to generate the data shown in Tables 1-3?**
>
> As you correctly noted, we include results under four domain orders in Appendix D.3. Tables 1–3 report results using Order N →B →W →D, which is also used in Section 4.
>
> We will explicitly state this in the main text to improve Clarity.
>
> ---
>
> **Q11. What is the model used in section 6? Add more details for the experiments and streamline such discussion with the main text.**
>
> As noted in Appendix C, we use a self-supervised version of the base ViT model as our default backbone.
>
> We will incorporate this information and additional experimental details from Appendix C into the main text to enhance Clarity.
>
> ---
>
> **References**
>
> [a] A tail-index analysis of stochastic gradient noise in deep neural networks. ICML 2019.
>
> [b] Hausdorff dimension, heavy tails, and generalization in neural networks. NeurIPS 2020.
>
> [c] The heavy-tail phenomenon in SGD. ICML 2021.
>
> [d] On the heavy-tailed theory of stochastic gradient descent for deep neural networks. arXiv preprint 2019.
>
> [e]  Stochastic Gradient Descent as Approximate Bayesian Inference. JMLR 2017.
>
> [f] Stochastic Gradient Descent Performs Variational Inference, Converges to Limit Cycles for Deep Networks.  ITA 2018.
>
> [g] A Simple Baseline for Bayesian Uncertainty in Deep Learning. NeurIPS 2019.

---

> ### Comment · Reviewer_ePCh · 2025-08-06
> **Thank you for the rebuttal**
>
> I'd like to thank the authors for their careful rebuttal, I am generally satisfied by their answers and the modest/realistically stated contributions of this paper. I keep my score and root for its acceptance :)

---

> > ### Author Response · Authors · 2025-08-08
> >
> > We sincerely appreciate your thoughtful response and your support for our work.

---

### Decision · Program_Chairs · 2025-09-17

**Decision:**

Accept (poster)

**Comment:**

The authors study learning dynamics during test-time adaptation, and empirically find that during successful test-time adaptation (TTA), the variance of the gradients follows a log-normal distribution. From here, the authors analyse that deviations from this empirically observed "sweet spot", model performance degrades (in TTA, often referenced to as model collapse) -- as the authors note, "The log-variance portrait thus provides a statistically grounded diagnostic signal for evaluating adaptation quality". This phenomenon is validated across artificially generated noise domains (Noise, Blur, Weather, and Digital). The authors consider some (DeYO, ROID) methods for sample-filtering and note that these methods incourage to stay close to the desired distributional properties.

Based on this observation, the authors propose an algorithm, SIGMA, which is an addition to existing test-time adaptation algorithms and aids with stable adaptations in scenarios where the algorithms would otherwise collapse. On ImageNet-C, the authors demonstrates that SIGMA is able to improve the performance of various methods and rescues the performance of TENT which would otherwise fail to adapt beyond the baseline level. For longer term adaptation, SIGMA can complement approaches like ROID and improves longer term adaptation performance on repeated ImageNet-C.

Reviewers recommend acceptance based on the interesting and convincing empirical observation and study of IG distributed variance (ePCh, TMwY, ryf8). Reviewer Popp raised justified concerns about the positioning within the literature studying learning dynamics of TTA, and the authors provided a suggested revision to their paper in the discussion phase. Concerns that the paper needs to provide more theoretical motivation were discussed and reviewers did not reach an agreement on whether additional theoretical justification is required.

Based on the discussions, and my own reading of the manuscript, I believe that this paper is a great example how an empirically and rigorously tested finding about model behavior can guide algorithm design. While I agree with reviewer Popp that additional theoretical investigation and and explanation for this behavior would be a valuable addition, such a theoretical investigation is beyond the scope of the current study (and would have made the difference to proposing the paper as a spotlight).

In general, I recommend acceptance, provided the authors make the following updates to their manuscript:

- It should be very clear that the paper discovers an empirical law and uses this to build an adaptation method. The authors might consider to tune down the presentation of *Theorem* 1, which on the first read seems misleading and like a central piece of the work, while it is in fact mostly incremental to Särkkä & Svensson [50]. This could be stressed more, also in the main paper, to clarify the contribution.
- Comparisons to additional state of the art methods
- More discussion on related work as provided during the rebuttal
- Additional benchmark datasets as provided during the rebuttal